# Smart Rural Communities: Action Research in Colombia and Mozambique

Igor Calzada [1,2,3,4,5]

1 Wales Institute of Social and Economic Research and Data (WISERD), Social Science Research Park (Sbarc/Spark), School of Social Sciences, Cardiff University, Maindy Road, Cathays, Cardiff CF24 4HQ, Wales, UK; calzadai@cardiff.ac.uk; Tel.: +44-7887661925

2 Fulbright Scholar-In-Residence (S-I-R), US-UK Fulbright Commission, Unit 302, 3rd Floor Camelford House, 89 Albert Embankment, London SE1 7TP, England, UK

3 Urban Transformations ESRC & Future of Cities Programmes, COMPAS, University of Oxford, 58 Banbury Road, Oxford OX2 6QS, England, UK

4 Ikerbasque, Basque Foundation for Science, Plaza Euskadi 5, 48009 Bilbao, Spain

5 Faculty of Social Sciences and Communication, University of the Basque Country, Barrio Sarriena s/n, 48940 Leioa, Spain; igor.calzada@ehu.eus; Tel.: +34-630752876

**Abstract:** This article contributes to the ongoing discussion on rural development programs aligning with the Sustainable Development Goals (SDGs) in the Global South. The research question examines how the Smart Rural Communities (SRC) framework can support the SDGs as an international cooperation model. The article presents findings from fieldwork action research including a critical analysis of the hegemonic discourse on smart cities and provides final recommendations. Additionally, it introduces the concept of SRC. The fieldwork action research was conducted in post-conflict rural areas in Colombia's Bolivar region and remote settlements in Mozambique's Cabo Delgado province. Led by Ayuda En Acción in collaboration with co-operatives such as Mundukide Foundation, Alecop, and LKS from Mondragon Co-operative Corporation, these interventions aimed to engage local communities through Living Labs. They utilized Information and Communication Technologies (ICT) and social innovation to promote the well-being of rural residents. The article comprises (i) an introduction; (ii) a literature review; (iii) a presentation of the fieldwork action research in Colombia and Mozambique; (iv) findings from a SWOT analysis and policy recommendations for SRC Living Labs; (v) conclusions addressing the research question. The SRC framework offers valuable insights for international cooperation models striving to achieve the SDGs in the Global South.

**Keywords:** smart rural communities; action research; SDGs; agenda 2030; Global South; living labs; social innovation; smart villages; smart cities; data cooperatives; data sovereignty

## 1. Introduction: Smart Rural Communities and Sustainable Development Goals in the Global South

The Sustainable Development Goals (SDGs) adopted by the United Nations aim to address the world's most pressing challenges by 2030, including poverty, hunger, and inequality, while ensuring environmental sustainability. In recent years, there has been a growing focus on rural development programs in the Global South that align with the SDGs, recognizing the significance of uplifting marginalized communities and promoting equitable development. Despite substantial efforts and investments, many rural communities in the Global South continue to confront various challenges such as limited access to basic services such as healthcare and education, as well as constrained economic opportunities. Moreover, rural areas often face a data and digital divide which hampers their participation in the digital economy and information society. Social exclusion further exacerbates these difficulties, perpetuating inequalities within and between communities.

The challenges faced by rural communities in the Global South are multifaceted and require a deeper examination to address the complex issues they face. Limited access to

basic services, including healthcare and education, remains a significant challenge. Without adequate healthcare infrastructure and educational opportunities, rural communities struggle to meet their fundamental needs and improve their livelihoods. Constrained economic opportunities are another crucial challenge faced by rural areas, with limited access to markets, financial services, and employment opportunities, hindering economic growth and development. The lack of investment in rural economies perpetuates poverty and inequality, exacerbating the disparities between rural and urban areas. In addition to these challenges, rural communities often experience a digital divide, further hampering their participation in the digital economy and broader information society. Inadequate internet connectivity, lack of digital skills, and limited access to technology impede their ability to access online services, engage in e-commerce, and benefit from digital innovation and opportunities for socio-economic advancement. These challenges contribute to social exclusion within and between rural communities. Marginalized groups, including women, indigenous populations, young people, and ethnic minorities, are particularly vulnerable, facing limitations in accessing resources, decision-making processes, and social networks. Social exclusion perpetuates inequalities and hinders the overall development and well-being of rural communities. To address these issues, comprehensive strategies are needed that focus on improving access to essential services, promoting inclusive economic development, bridging the digital and data divide, and fostering social inclusion. Efforts should be made to strengthen healthcare infrastructure, enhance educational opportunities, invest in rural economies, expand digital connectivity, and empower marginalized groups through targeted interventions and policies. Moreover, it is crucial to ensure that rural communities have a voice in decision-making processes and that their perspectives and needs are considered in policy formulation and implementation. Collaborative efforts involving governments, civil society organizations, NGOs, academia, and private sector entities are necessary to drive sustainable change and promote equitable development in rural areas. By recognizing and addressing the complex challenges faced by rural communities in the Global South, more inclusive, resilient, and sustainable frameworks can be created where no one is left behind.

This article aims to contribute to the ongoing discussion on rural development programs and their alignment with the SDGs in the Global South, both before and after the pandemic. By examining the successes, limitations, and lessons learned from existing initiatives, it seeks to shed light on effective strategies for addressing the multifaceted challenges faced by rural communities. Furthermore, although the fieldwork action research was conducted in 2017, before the pandemic, the article aims to retrospectively reflect on the impact of the COVID-19 pandemic on rural development efforts, considering the findings and evidence from the fieldwork action research conducted in Colombia and Mozambique.

By providing insights into the intersection of rural development programs and the SDGs, this article aims to inform policymakers, practitioners, and researchers working in the field of sustainable development. Understanding the nuances and complexities of rural development can enhance our collective efforts to create more inclusive, resilient, and sustainable societies in the Global South and beyond.

This article introduces a new experimental policy intervention model called 'Smart Rural Communities' (SRC), challenging the three main joint pillars of the SDGs policy (cities, villages, and citizens) that have become popular in the Global North. The article presents fieldwork action research conducted in post-conflict rural areas in Colombia's Bolivar region and remote rural settlements in Mozambique's Cabo Delgado province. The research was directed by the author of this article (while a Senior Researcher at the University of Oxford; https://www.ox.ac.uk/news-and-events/find-an-expert/dr-igor-calzada, accessed on 1 April 2023) and led by the NGO Ayuda En Acción (based in Madrid, Spain) in collaboration with Mondragon Co-operative Corporation (MCC) co-operatives (based in the Basque Country, Spain) including Mundukide Foundation, Alecop, and LKS. This intervention aimed to engage local communities through the incorporation of information and communication technologies (ICTs) to promote the well-being of rural residents.

The fieldwork action research was undertaken sequentially in both locations in the Global South [1,2]. The intervention aimed to directly engage with rural dwellers in both locations rather than compare the cases, allowing for the modeling of SRC. The article presents the action research methodology used to build Living Labs around rural communities with unique cultural traits. Through insightful intervention, it shapes the experimental model entitled SRC based on qualitative fieldwork action research [3,4]. The fieldwork action research was conducted before the pandemic, between June and August 2017, to explore the potential technological empowerment among rural dwellers in vulnerable communities and remote settlements [5]. Rural areas in both countries faced significant challenges in accessing basic services including education and healthcare. Internet access was limited in rural areas in Bolivia, particularly in indigenous communities, making it difficult to provide virtual educational services. Similarly, in Mozambique, access to education was limited in rural areas, with significant disparities between urban and rural areas in terms of school attendance and educational outcomes. Efforts have been made to promote rural development and the use of technology in both countries. For example, in Colombia, there were initiatives to promote the use of ICTs in rural areas to improve access to education and healthcare services. In Mozambique, the government developed a national strategy for ICT development aimed at promoting digital inclusion including in rural areas. These efforts align with the concept of SRC, which aims to leverage the potential of technology to improve the well-being of rural communities. Moreover, the pandemic has further exacerbated these needs and misalignments between ICTs, datafication, and rural communities.

Against this backdrop, the role of technology in rural development has gained prominence in recent years, with digital innovations and the increasing availability of ICTs. Integrating technology into rural development programs has the potential to improve access to essential services, enhance economic opportunities, and promote social inclusion. However, the use of technology in rural development is often dominated by the discourse of Smart Cities (SC), which tends to focus on urban environments and neglects the needs and realities of rural communities. This article provides a critical analysis of the hegemonic discourse surrounding SC and its limitations in addressing the challenges of rural development. Furthermore, this article revolving around SRC as an experimental intervention model aims not only to challenge the postcolonial SC rationale imposed from the Global North but also to establish an ad hoc contextualized version for rural, vulnerable, remote, and communities based on Living Labs [6–8]. It introduces the concept of SRC, leveraging the potential of technology while addressing the specific needs and challenges of rural communities. The article includes an introduction, literature review, presentation of the fieldwork in Colombia and Mozambique, findings through a SWOT analysis, and final policy recommendations for the SRC framework.

Hence, the research question of this article is whether a new development model for international cooperation is feasible by experimenting through Living Labs with SDGs in the Global South. This research question was presented and discussed at the Summer School SRC, which took place on 12–13 September 2019 in San Sebastián, Basque Country, Spain (https://www.uik.eus/es/node/5812/pdf, accessed on 1 April 2023). This would enable digital transformational processes to be implemented as grassroots innovation in collaboration with remote and rural community dwellers.

## 2. Literature Review: The State-of-the-Art on Smart Cities (SC), Smart Villages (SV), and Smart Rural Communities (SRC)

In the context of rural development and the pursuit of SDGs, there is an increasing recognition of the concept of SRC. SRC encompasses the application of innovative technologies and digital solutions to address the unique challenges faced by rural areas, promoting inclusive and sustainable development. While the focus on smart urbanism has been prominent in the Global North, it is important to shift the narrative and consider the potential of smart solutions in overcoming rural conditions in the Global South. In the

Global South, the challenges faced by rural communities are diverse and require context-specific interventions. The SDGs provide a framework that highlights the importance of rural development as a thematic area, acknowledging the interlinkages between rural development, agriculture, land, and other related issues. Efforts to develop SRC in the Global South are rooted in the understanding that technology alone is not a panacea, and a comprehensive approach is needed to address the complexities of rural contexts. The focus should shift towards understanding the significance of communities, sustainability, and the specific needs and aspirations of rural dwellers. By combining technological solutions with community-driven approaches, the aim is to foster sustainable development, enhance quality of life, and bridge the rural–urban divide. While SRC offers immense potential for inclusive and sustainable development, it is crucial to avoid replicating the one-size-fits-all approach often seen in smart urban initiatives. Instead, it is important to engage with the complexities and nuances of rural areas in the Global South, considering local contexts, cultural values, and the diversity of rural communities. By doing so, smart rural solutions can be tailored to address specific challenges such as limited access to services, constrained economic opportunities, and social exclusion. By embracing the concept of SRC in the Global South, integrating technological advancements with community-centric approaches, and aligning these initiatives with the SDGs, there is an opportunity to create sustainable and resilient rural environments that empower individuals, enhance livelihoods, and contribute to the overall well-being of communities.

In the Global North, corporate SC rhetoric portrays technology as a catch-all solution for social, economic, and environmental urban issues [9–14]. A politicized point of departure encourages tabula rasa interventions and replaces as hegemonic the normative rationale behind the notion that smart urbanism should overcome the rural conditions for their dwellers. Without engaging with complexities, technologies may not solve but rather perpetuate existing issues in the Global South [15–18].

SC debates tend to focus on how well technology serves the city toward predetermined goals [19–22]. Much of the corporate literature emphasizes how big data and the evolution of hardware (the Internet of Things) can contribute to more transparent governance and effective monitoring of city infrastructure and services. In developmental contexts, technology is often seen as an enabler, a positive force that can be harnessed for socio-economic development. However, there are two main issues with this interpretation: (i) There is a tendency to view innovation as a force that exists outside of human interaction, knowledge, and experience, driven solely by experts; (ii) the assumption that the 'old' will be replaced by the 'new' with the broadened availability of technological tools may not necessarily come true [23].

This article argues that a revised perspective that engages with rural dwellers is required. In this regard, the intervention in the Global South initially revealed that socio-technical processes manifest spatially as the relationships between the material (technology, infrastructure, and natural systems) and human agency (social action, planning, and culture) evolve. This represents an interaction between technological innovation and the construction and appropriation of social innovation processes. Moulaert and MacCallum define social innovation as 'innovation in social relations based on values of solidarity, reciprocity and association' [24] (p. 1). The scope of this article is to explore how social innovation can help emancipate rural communities in the Global South through digital transformations [25–27].

COVID-19 was spreading rapidly and its tragic aftermath showed that the world is highly interconnected. Acknowledging the particularities of the Global South in relation to the Global North is necessary to solve a great number of problems [28]. Shockingly, COVID-19 made all world citizens pandemic citizens, sharing the same fear, uncertainty, and risks regardless of their location in the world [29–31]. However, it was unlikely that the pandemic crisis and its algorithmic disruptive vulnerabilities equally affected citizens in the Global South and the Global North. It became evident that the pandemic crisis forced the world into an algorithmic crisis, in which citizens' data could be used for unfair

or unethical purposes by governments or private companies. The proliferation of new emerging digitalization/datafication apps, including ChatGPT and Metaverse among others, only served to confirm this early intuition. Above all, and while considering the digital risks, Living Labs are among the various resilience strategies worth considering addressing the aftermath of COVID-19 through social innovation [32–34].

Against this backdrop and in line with the SDGs policy framework, this article introduces a new experimental policy intervention model called SRC. In contrast to the prevailing trend of SC policy in the Global North, this model challenges the three main pillars of cities, villages, and citizens [35–41]. SRC emerged as a result of extensive research and policy findings derived from a fieldwork action research project conducted in various rural, vulnerable, and remote communities in the Global South in 2017. Specifically, the research focused on post-conflict areas in Colombia (Latin America) and scattered and newly developed regions in Mozambique (Africa). The insights and data obtained from this qualitative fieldwork action research informed the development of the SRC intervention model.

Hence, this article interweaves the state-of-the-art interrelated concepts, such as SC [42], SV [43], Living Labs [44], and action research, along with their impact on the implementation of the SDGs. It focuses on three main aspects: (i) the feasibility of technology, (ii) the role of politics and power relations within communities, and (iii) the self-capacity of communities to develop their locally driven entrepreneurial model based on (data) co-operativism [45,46].

The starting point of this article is the recognition that new technologies or smart technologies coexist with 'older' versions, and this relationship is strongly influenced by structural factors [47,48]. The article argues that understanding these digital transformational processes is crucial to guide investments and interventions in SC technology that are meaningful and contextually relevant for Colombia and Mozambique. Following the introduction and literature review, which provide the foundation for this perspective and present the main research question, the article proceeds to justify the research intervention in the subsequent section. It does so by describing the methodology of the fieldwork action research conducted through Living Labs in Colombia and Mozambique. Finally, the article concludes by presenting policy recommendations for both countries and discussing the future implications in light of the current post-COVID-19 context in remote and rural communities in the Global South.

Based on the provided research results, there is a growing interest in the practice of participatory approaches to developing ICTs for rural agricultural communities [49]. The Enabling Rural Innovation (ERI) approach is an innovative action research approach that aims to strengthen social and entrepreneurial capacity in rural communities. The approach focuses on fostering community-based capacity for the inclusion of rural women, young people, and the poor in analyzing market opportunities. The use of action research has been found to help the farming community adopt ICT-based solutions for agriculture, which, in turn, contributes to problem-solving and assists in decision-making by identifying technical and agricultural needs.

SRC is an emerging field of interest, highlighting the importance of understanding the role of digital technology in rural development [50]. The concept of the smart society is a global movement that underscores the advancements in digital technology and the inherent contradictions it brings. However, existing studies on the smart society predominantly focus on the application of technology to support human activities, particularly in urban areas or simply on SC [51]. The understanding of how technology impacts rural communities remains limited. The article's findings revealed that rural communities have the ability to access and leverage external resources to create value within their communities. Moreover, interactions between rural and urban communities foster a learning process and generate innovative ideas. One such idea is the use of digital technology to address challenges in rural areas. Participatory design methods, including action research, can be employed to educate rural individuals in ICT. PunCar Action, a volunteer program in

Taiwan, exemplifies this approach where ICT educators travel to rural communities and offer courses on digital technology usage. The participatory design proves to be an effective strategy for teaching ICT and Web 2.0 skills, facilitating the co-creation of community blogs, and sustaining intrinsic motivation to use Web applications. PunCar Action presents an innovative bottom-up intergenerational ICT education model with wide reach, capable of boosting the confidence of rural residents in utilizing ICT.

SV is an increasingly important area of interest for scholars, practitioners, rural areas, and communities [52]. Rural areas are significantly affected by spatial vulnerability, the digital divide, depopulation, and population aging. Marginalized populations are striving for collective well-being, entrepreneurship, digital literacy, social inclusion, and local development within the context of SV. In Greece, there have been limited interventions in SV, primarily focusing on social innovation, entrepreneurship, and the use of ICTs to enhance the quality of life in rural areas. Innovation, knowledge, growth, and management play crucial roles in rural smart planning.

In conclusion, the utilization of action research approaches and ICTs can support rural communities in adopting smart solutions for agriculture and rural development [1–4]. Participatory design methods can be applied to educate rural individuals in ICT. SV is an increasingly significant area of interest for scholars, practitioners, and rural areas and communities [53].

SV is a concept aimed at enhancing traditional rural aspects through digital transformation. The literature on SV [54–59] covers various topics, including application development, management of Information Technologies, strategy, and societal implications. It also addresses challenges and pitfalls in rural development and suggests ways to overcome them. The concept of SV is relatively new in EU decision-making and policy, necessitating the involvement of multiple stakeholders ranging from rural residents to decision-makers to identify the strengths, threats, opportunities, and weaknesses to a specific rural area. The literature also emphasizes the importance of ecosystem sustainability as a fundamental requirement for all development plans, whether SC or SV. The state-of-the-art and the literature review indicate that the successful implementation of emerging and existing technologies in rural development relies on the active participation and empowerment of residents. Smart solutions alone, without the will and decision-making of rural dwellers, may not yield the desired outcomes. Furthermore, both SV and SRC focus on improving living conditions and the quality of life in rural areas, going beyond the technology-centric approach associated with SC [41,60]. The core idea behind SV and SRC is not to seek ICT-driven solutions but to create conducive living conditions for a fulfilling life and increased engagement of the rural population [53]. Hence, SV and SRC share a similar approach, although SRC specifically emphasizes the creation of a living ecosystem through the direct involvement and empowerment of rural dwellers, achieved through Living Labs.

It is noteworthy, however, that the SV concept has been implemented for a relatively short period and is particularly linked to other terms such as 'village renewal' or 'sustainable development' in the EU context [61]. The latter aligns closely with the SDGs, which is why this article contributes to the ongoing discussion on rural development programs and their alignment with the SDGs in the Global South. Therefore, SRC can be viewed as a critical response to the Global North-driven SC and as an extension of the SV concept with a strong emphasis on 'community development' and the SDGs in the Global South.

Furthermore, SV is a relatively new concept among EU decision-makers and policy-makers, emerging from years of debates surrounding economic and territorial inequalities, social exclusion, diversification of certain areas, the gradual reduction of agricultural activities, and the intersection of cohesion, regional, and common agricultural policies [62]. The concept of SV encompasses the preservation of villages and their inhabitants, the protection of cultural heritage, and the utilization of local resources to address contemporary challenges. The SV approach is an ICT-conscious integrated strategy that offers sustainable solutions to the issues faced by rural communities, such as depopulation, diminishing services, and inadequate infrastructure in areas with aging populations. By implementing

the SV concept, it becomes possible to tackle the key challenges encountered in rural areas including depopulation, an aging society, climate change, increasing food demand, environmental degradation, peripheralization, low income among rural populations, and the impact of the COVID-19 pandemic. The SV approach provides an alternative perspective on enhancing the quality of rural life and appears to align with the evolving EU policy direction. However, it necessitates more tailored tools and instruments, both at the EU level and within national/regional contexts. Regional and local governments also have a crucial role to play in this process [63].

Consequently, in this article, SRC is presented as a model that can be implemented endogenously from within the Global South, driven by action research and conducted through Living Labs.

### 3. Methods and Materials: Action Research Fieldwork in Colombia and Mozambique

Action research, as defined by Lewin [3] (p. 35), is 'transformative research on the conditions and effects of various forms of social action, leading to social action that employs a spiral of steps, each consisting of a cycle of planning, action, and fact-finding to assess the outcomes of the action'. Action research aims for transformative change through the integrated process of taking action and conducting research and interconnected by critical reflection [1,2,4].

This section presents the fieldwork action research conducted in rural and remote communities in post-conflict areas of Colombia (Latin America) and newly developed areas in Mozambique (Africa) (Figure 1). It provides qualitative data to shape an intervention model called SRC. The project aims to challenge the prevailing SC approach from the Global North and establish a context-specific version for rural communities in strategically targeted locations in the Global South. Addressing concerns regarding the appropriation of action research in the Global South, this article demonstrates how action research, conducted through Living Labs, can operate in collaboration with rural dwellers, respecting their environment and enabling them to lead the intervention process. Drawing on post-colonial literature on SC, the article emphasizes the importance of avoiding methodological nationalism in action research [64–66].

The project was led by the NGO Ayuda en Acción (Aid-in-Action), based in Spain, which implemented and applied the resulting strategic outcomes internationally across their territorial development areas and branches. While the NGO has been actively engaged in international aid efforts, this project enhanced the NGO's potential strategy by incorporating the 'smart' utilization of ICT, energy, mobility, education, health, gender, and governance advancements in conjunction with a participatory and experimental approach through Living Labs. The project aimed to update the operational framework of the NGO Ayuda en Acción as an international development and humanitarian aid organization.

The action research design consisted of three phases (Figure 2): (i) state-of-the-art analysis [41,53,62], (ii) fieldwork action research [1], and (iii) modelization [67]. The fieldwork research employed three action research techniques: (i) visual ethnography, (ii) in-depth interviews, and (iii) Living Labs, supplemented by focus groups. This project demonstrated a policy commitment to revitalizing the strategic and operational intervention models of the NGO Ayuda en Acción by incorporating valuable lessons learned from the field, encompassing both infrastructure development and community capacity building. The project sought to establish strategic alignment with supranational institutions in this field such as the Inter-American Development Bank (BID), the European Union (EU), UN-Habitat, and the OECD, among others.

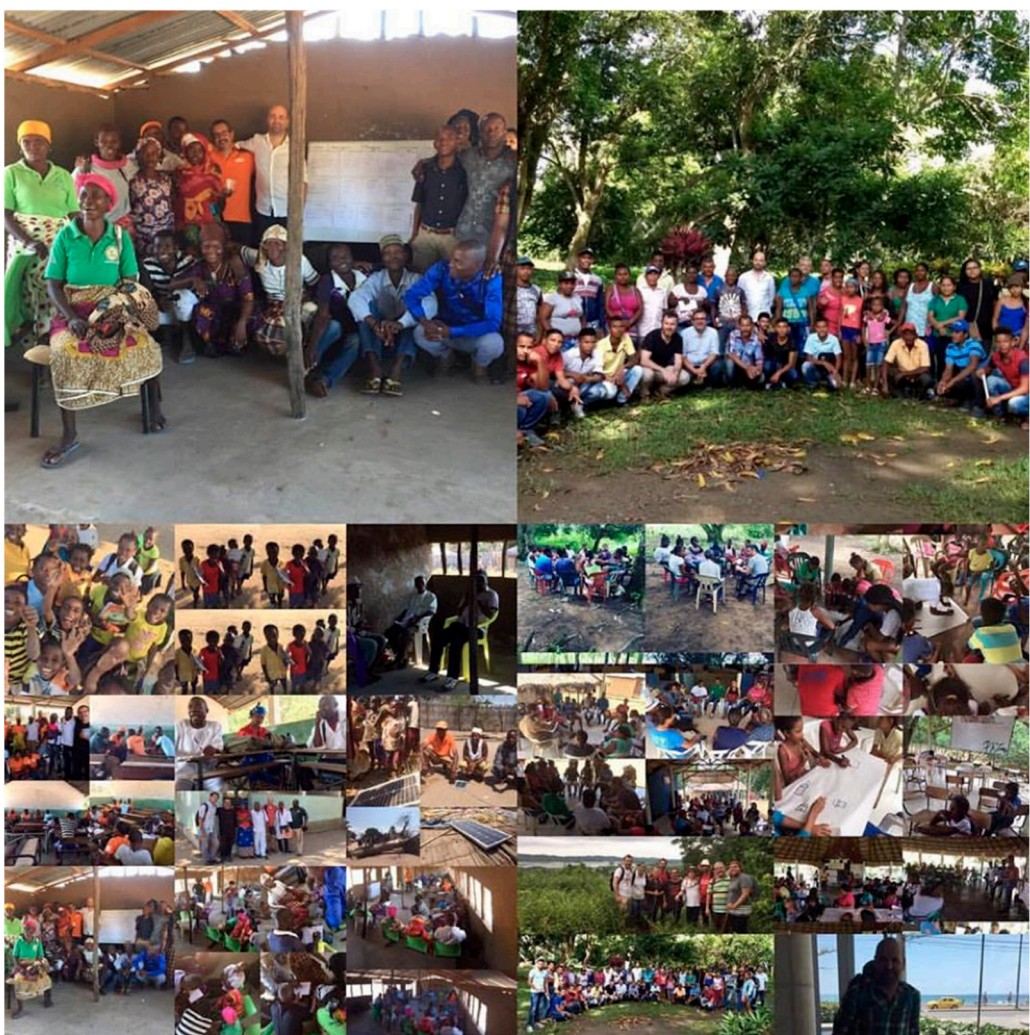

**Figure 1.** Mozambique and Colombia Fieldwork Action Research: SRC.

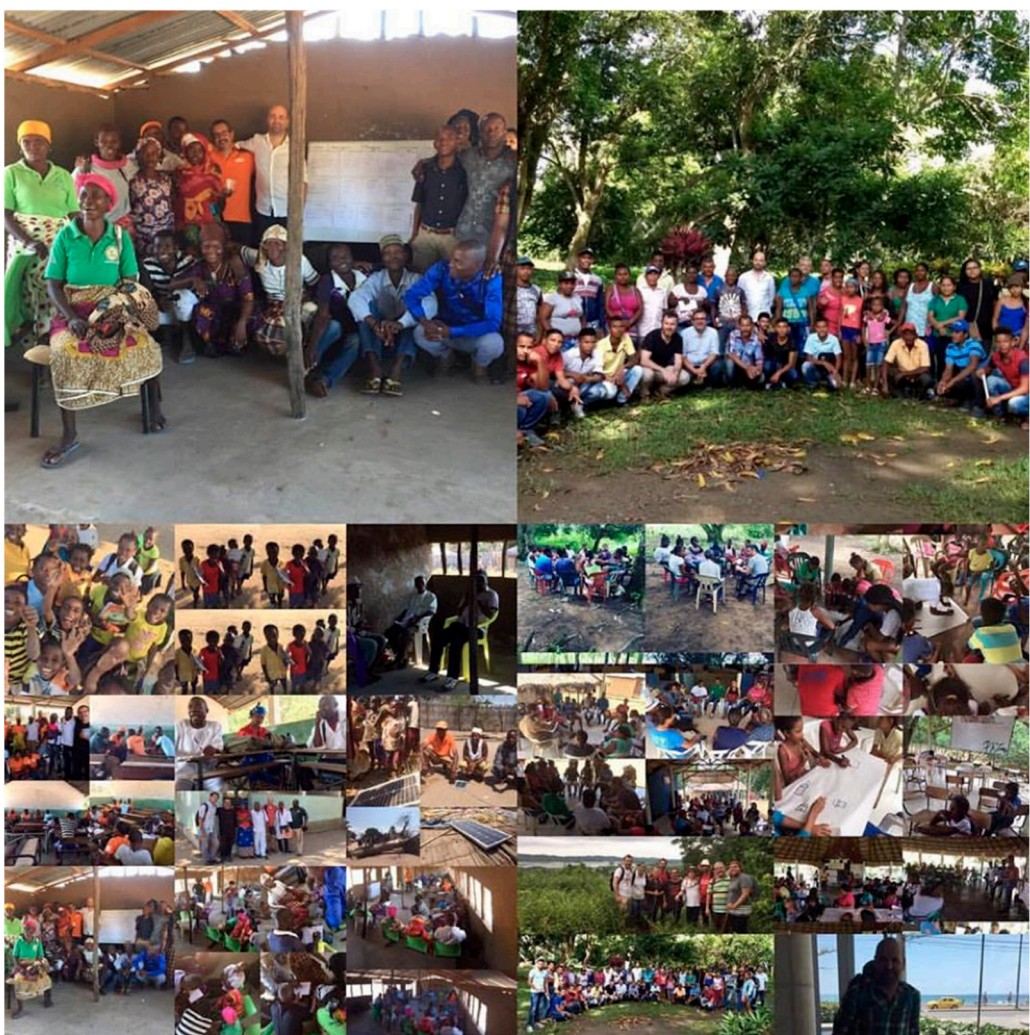

**Figure 2.** Action research design [41].

In the two aforementioned cases, residents from rural and remote communities actively participated in fieldwork action research. The project aimed to understand their

perspectives and explore ways to enhance their daily lives by leveraging incremental technological advancements. The fieldwork sought to identify stakeholder groups and examine their interconnectedness in promoting community empowerment through smart strategies and cooperative socio-economic development at the grassroots level.

These cases were conducted in conflict-affected regions, each with its own unique circumstances. The communities in Bolivar (Colombia) involved in the project were deeply affected by the consequences of the conflict between the *Fuerzas Armadas Revolucionarias de Colombia* (FARC) and the Colombian Government. On the other hand, the community selected in Pemba (Mozambique) experienced significant population displacement due to insurgency. Both cases were purposefully chosen by the NGO Ayuda en Acción due to their profound impact on rural residents and the potential they offered for experimentation and social innovation.

In Bolivar (Colombia) (Figure 3), the three selected communities were directly affected by the conflict between paramilitaries (*paracos*) and the FARC guerrilla. The project took place in the post-conflict context of Bolivar following the peace agreement between the Colombian government and the FARC rebel group, which was announced in late 2016.

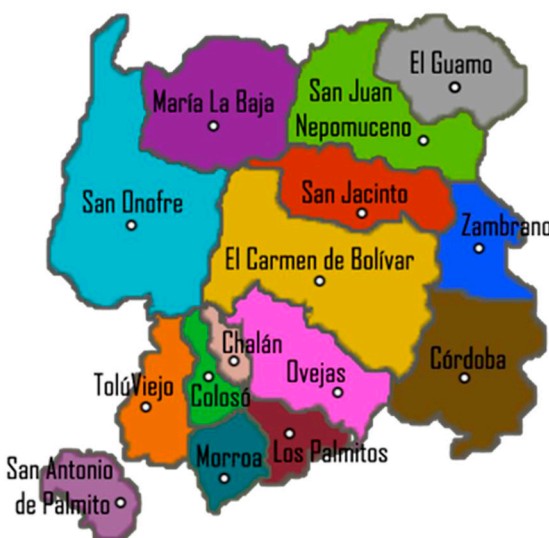

**Figure 3.** Colombia: Bolivar.

The peace agreement marked a significant milestone in Colombia's history, aiming to bring an end to the country's 52-year civil conflict that resulted in the loss of numerous lives and the displacement of millions of people, particularly in post-conflict areas such as Bolivar. The agreement addressed crucial issues such as land reform, political participation, illicit drugs, and the rights of victims.

However, the post-conflict period in Colombia has not been without challenges. One notable issue has been the ongoing violence against social leaders, which has persisted despite the peace agreement. In the aftermath of the agreement, territorial disputes and struggles for control over illicit markets have emerged involving armed non-state actors (ANSAs) who were not party to the agreement. Additionally, several dissident groups have emerged from within the FARC, further complicating the post-conflict situation.

In Pemba (Mozambique) (Figure 4), the project focused on seven communities that were directly affected by various struggles. By 2017, the region of Pemba in northern Mozambique had already witnessed a significant increase in population due to the ongoing insurgency and violence in the area. Pemba, a city with a population of just over 200,000 in 2017, had experienced substantial displacement of people as a result of the insurgency. Nearly 690,000 individuals had been displaced since the insurgency began, with many seeking refuge in Pemba. During this time, the violence in northern Mozambique led to extensive destruction. Human rights groups documented incidents of killings, beheadings, and kidnappings carried out by the militant groups operating in the region. These attacks

caused deaths and instilled widespread fear among the local population. In May 2021, reports emerged of people desperately attempting to flee Pemba and the surrounding area following a jihadist attack. Security forces had set up roadblocks, effectively trapping thousands of people. In their desperation to escape, some individuals had to pay bribes to be allowed to leave the town.

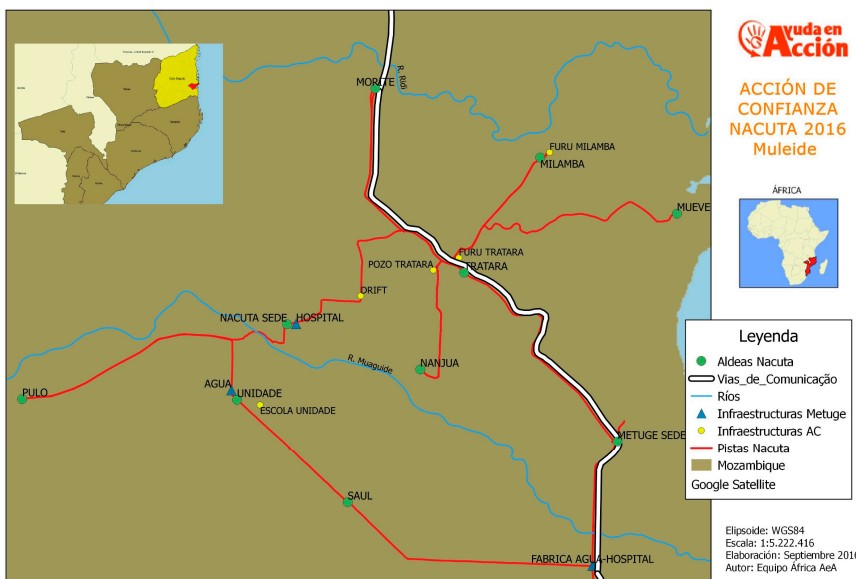

**Figure 4.** Mozambique: Pemba.

Given that fieldwork action research was conducted retrospectively in 2017, it is worth noting that regional conflicts were ongoing both before 2017 in the case of Colombia and after 2017 in the case of Mozambique. The decision to select post-conflict areas was a deliberate choice in designing the fieldwork action research. As a result, SRC was tested in real and challenging environments with respect to achieving the SDGs. Considering the presence of regional conflicts and political unrest prior to the intervention in Colombia and unfortunately persisting in Mozambique after the intervention, it becomes difficult to measure the impact of these micro interventions at the macro and meso levels [68,69]. Therefore, conducting a cause-and-effect analysis regarding changes in the behavior of the communities before (in the case of Bolivar, Colombia) and after (in the case of Pemba, Mozambique) goes beyond the scope of this article. The primary objective of this intervention was to experiment and test Living Labs as potential mechanisms for community empowerment by exploring the use of ICT [70–73].

Two groups of stakeholders were given particular consideration throughout the intervention process: millennials and women. In the case of the first group, particularly in Colombia, it posed a significant challenge as young entrepreneurs were returning to their villages after an extended period of absence due to the conflict. In Mozambique, the project explored the role and potential involvement of women in collectively run agricultural associations. The overarching goal of the project was to inspire local residents and natives to initiate their own entrepreneurial ideas while receiving support from experts and technical professionals. Participants engaged in various action research activities within their communities, collectively reflecting on their current and future living and working conditions. By focusing on these target groups, the SRC project aimed to empower them in the community's decision-making processes, fostering community empowerment through the opportunities provided by smart strategies and cooperative socio-economic development from the ground up.

Furthermore, this article centers around the SRC as an experimental intervention model. Its aim is not only to challenge the postcolonial rationale of SC imposed from the Global North but also to establish a context-specific version tailored to rural, vulnerable,

and remote communities. The SRC critically examines the 17 SDGs and the New Urban Agenda-Habitat III coordinated by UN-Habitat through the lens of the two aforementioned experimental action research fieldwork processes.

The SRC experimental intervention model has been scientifically led since 2016 by the author of this article, who held the position of Senior Researcher at the University of Oxford, in close collaboration with the NGO Ayuda en Acción (Aid-in-Action). Initially, the goal of SRC was to reformulate the intervention strategy of this NGO by incorporating the "smart" utilization of ICT, energy, mobility, education, health, gender equality, youth and women entrepreneurship, and governance advancements, alongside a participatory and experimental methodology based on Living Labs. As a result, this article contributes to the reorientation of NGOs, such as Ayuda en Acción in Spain, as international organizations for development and humanitarian aid by embracing an experimental approach.

The design of the fieldwork action research focused on local residents in rural, impoverished, and remote communities referred to as 'rural citizens'. The research aimed to understand their perspectives and explore ways to enhance their daily lives through the implementation of technological advancements. Throughout the intervention process, particular attention and targeting were given to two groups of stakeholders: Millennials and women.

To achieve its objectives, the project utilized the concept of 'Living Labs', which is a participatory and experimental methodology aimed at exploring the meaning of rural Living Labs in the context of this specific project. The Living Labs approach facilitated the identification of stakeholder groups and the examination of their inter-dependencies, as well as the dynamics influenced by the community's past experiences, ultimately enhancing the overall capacity for community empowerment. Living Labs are collaborative environments where researchers, industry representatives, government entities, and communities collaborate to co-create, prototype, and test innovative solutions to societal challenges. These labs serve as real-world testing grounds, enabling researchers to gain valuable insights into user needs and behavior while providing communities with access to new technologies and services that can significantly improve their quality of life.

As shown in Figure 2, both case studies, Bolivar (Colombia) and Pemba (Mozambique), followed the same action research design.

1. The first methodological step involved conducting a literature review to establish the state-of-the-art around SC, SV, and SRC.
2. The second methodological step focused on organizing the action research fieldwork. This step included three techniques: Visual ethnography, in-depth interviews with rural dwellers, and convening the Living Lab. Visual ethnography involved capturing a large volume of videos and photos to understand the physical and material composition of the remote communities under examination. In parallel, in-depth interviews were conducted to identify community roles and development challenges for each community. The last step involved convening the Living Lab after collecting information. Living Labs were organized around four main areas: Energy, Production, Education, and Technology. Both case studies followed the same structure of the Living Lab, which facilitated finding connections between the cases and making slight comparisons. Figure 5 shows the outcomes of the Living Lab in Bolivar (Colombia) and Figure 6 shows the outcomes in Pemba (Mozambique). These outcomes were entirely generated by rural communities after a month-long iterative process of fieldwork action research. The outcome of the fieldwork action research achieved a high level of participation from the communities, with rural dwellers showing enthusiastic and supportive responses.
3. Finally, the fieldwork action research culminated in the modeling of the SRC as an intervention framework.

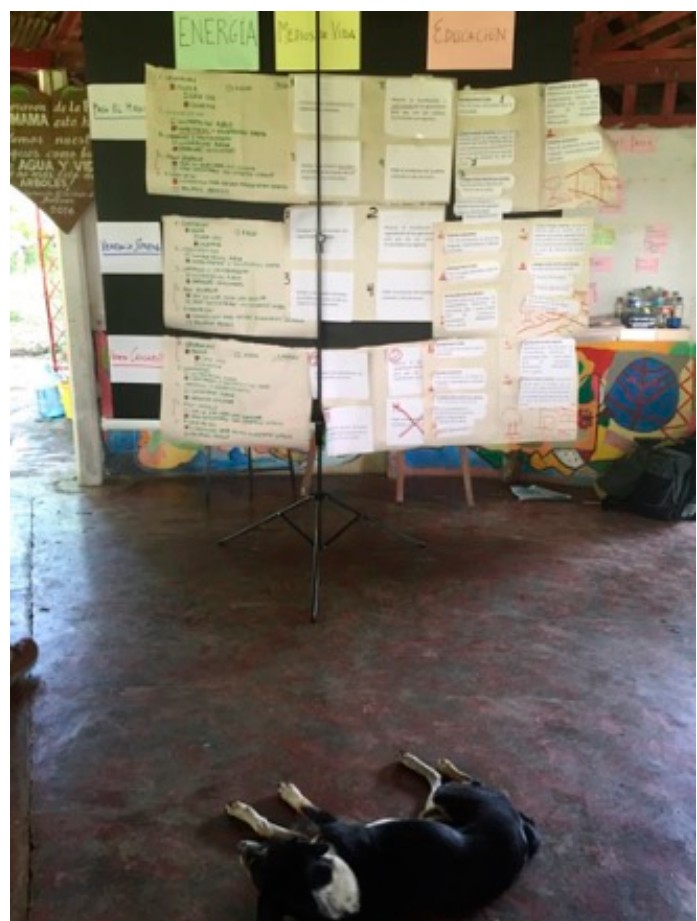

**Figure 5.** Colombia: Bolivar. Living Lab outcome.

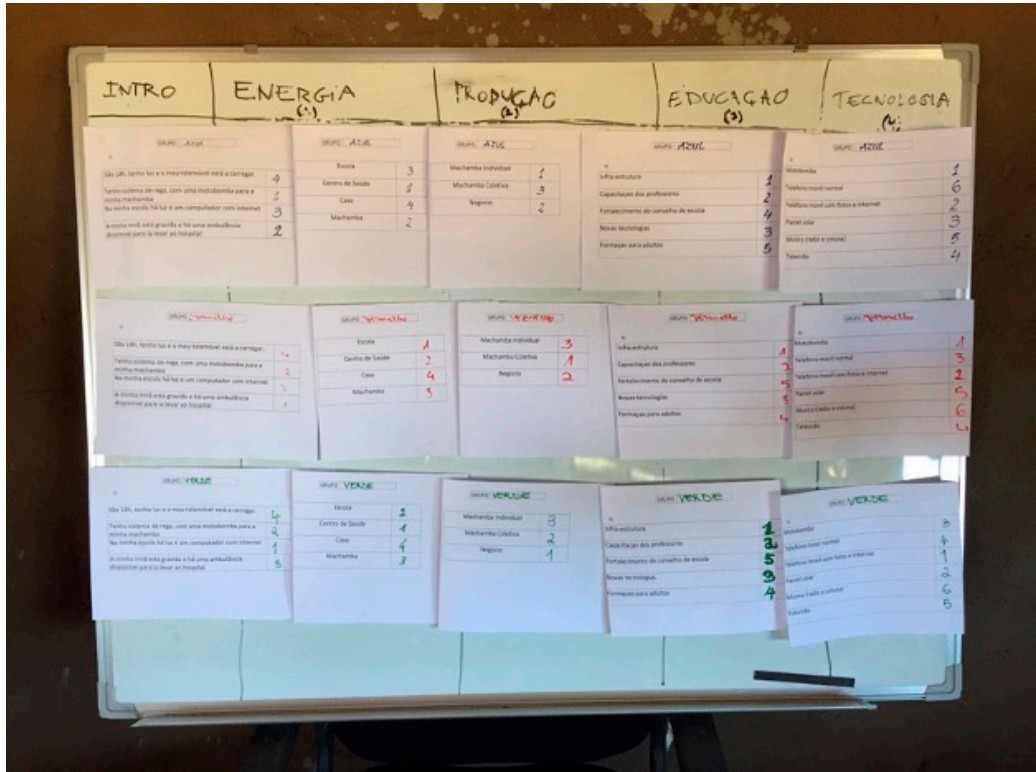

**Figure 6.** Mozambique: Pemba. Living Lab outcome.

The outcomes of step (i) are presented in the second section of this article, the outcomes of the fieldwork are presented in the third section, and the outcomes of the modeling are presented in the fourth section.

In the context of rural and remote communities in Mozambique and Colombia, Living Labs have been established to address specific challenges faced by these communities including limited access to healthcare, education, and transportation. These Living Labs focus on developing solutions that are locally relevant, sustainable, and scalable.

In Mozambique, Living Labs have been established in the provinces of Inhambane and Cabo Delgado to tackle healthcare and education challenges. The Living Lab in Inhambane aims to improve maternal and child healthcare by providing access to telemedicine services and developing health technologies tailored to local needs. Meanwhile, the Living Lab in Cabo Delgado focuses on enhancing access to education through the use of technology such as e-learning platforms and mobile apps.

In Colombia, Living Labs have been set up in rural and remote areas to address transportation, energy, and agriculture challenges. For instance, the Living Lab in Guajira is dedicated to developing sustainable energy solutions, such as solar panels and wind turbines to address the region's limited access to electricity. On the other hand, the Living Lab in Cauca focuses on improving agricultural productivity through the utilization of precision farming technologies.

Overall, Living Labs in rural and remote communities in Mozambique and Colombia serve as important platforms for promoting innovation and sustainable development, while improving the lives of local communities.

Through diverse research activities, including visual ethnography, in-depth interviews, and Living Labs, participants collectively reflected on their present and future living and working conditions across four key areas: Energy, Production, Education, and Technology (as shown in Figures 5 and 6). This allowed them to initiate their own entrepreneurial ideas with the support of experts and technical professionals specializing in energy, production, entrepreneurship, and education. Each professionally conducted sessions in their respective areas of expertise: Energy (LKS, Mondragon, Spain), Production (Mundukide Foundation, Mondragon, Spain), Education (Alecop, Mondragon, Spain), and Technology (University of Oxford, Oxford, UK).

In conclusion, the SRC project aimed to establish a context-specific version of the SC concept for rural communities in strategically targeted locations in the Global South. By incorporating the smart use of ICT, energy, mobility, education, health, gender equality, youth and women entrepreneurship, and governance advancements, along with a participatory and experimental methodology, the project aimed to enhance the strategies of the NGO Ayuda en Acción. Through its fieldwork action research process, the project sought to empower local communities, with a particular focus on *millennials* and women, thereby promoting community empowerment and sustainable development.

Action research is a problem-solving approach that combines research, action, and reflection to identify and address practical problems in real-world contexts. This approach was particularly useful for projects such as SRC that aimed to overcome barriers to sustainable development in rural areas. Here are some ways in which action research helped overcome barriers in the SRC project:

(i) Identifying barriers: Action research helped to identify the specific barriers that are preventing rural communities from adopting new technologies and practices. This included barriers related to access, affordability, and cultural norms.

(ii) Co-creation: Action research involved collaboration between researchers, community members, and other stakeholders to co-create solutions that were tailored to the specific needs of the community. This approach helped overcome barriers by ensuring that solutions are relevant, acceptable, and feasible for the community.

(iii) Testing and refinement: Action research involved testing and refining solutions in real-world settings to ensure their effectiveness and sustainability. This helped overcome barriers by identifying any implementation challenges and addressing them promptly.

(iv)  Scaling up: Action research helped identify strategies for scaling up successful solutions to other communities or regions. This helped overcome barriers by demonstrating the potential impact of the solution and encouraging wider adoption.

In the context of SRC, action research helped overcome barriers to sustainable development by identifying and addressing the specific challenges faced by rural communities, co-creating solutions tailored to their needs, testing, and refining these solutions in real-world settings, and scaling up successful solutions to other communities or regions.

## 4. Discussion: Findings and Policy Recommendations for SRC Living Labs

Given that action research is a research approach that involves actively working with a community or organization to identify and solve problems, it is a collaborative and iterative process that involves cycles of planning, action, observation, and reflection. In the context of SRC, action research was used to help identify the unique challenges and opportunities facing these communities and to develop and implement solutions that are tailored to their specific needs. For example, action research was used to develop strategies for improving access to high-speed internet, promoting economic development, and addressing environmental concerns. One of the key benefits of action research is that it involves the active participation of community members, which helps to ensure that the solutions developed are relevant and effective. Additionally, the iterative nature of the process allows for ongoing feedback and adjustment, which can help improve the outcomes of the research and implementation efforts. Overall, action research can be a powerful tool for helping to create smarter, more resilient, and more sustainable rural communities.

After conducting the literature review and arranging the fieldwork action research (which included visual ethnography, in-depth interviews, and convening two Living Labs, one for each case study), the action research team, consisting of the author of this article (as the scientific director of the project representing the University of Oxford, Future of Cities and Urban Transformations ESRC Programmes), the CEO and key staff from the NGO Ayuda en Acción, professionals from the three co-operatives (Mundukide Foundation, Alecop, and LKS) that directly co-designed the Living Labs, and participants of the Living Labs in Bolivar (Colombia) and Pemba (Mozambique) examined the findings and agreed upon policy recommendations. The decision-making process was not based on a common consensus as there were different viewpoints. The modeling process took more than one year to complete after the fieldwork action research on-site. However, the fieldwork action research, conducted through the three main techniques, was robust enough to generate policy recommendations that were ultimately embraced by the NGO Ayuda en Acción at the end of the process. The SWOT analysis presented in Tables 1 and 2 was used to establish common ground and reach policy recommendations. These SWOT analysis results were agreed upon by all the stakeholders involved in this project.

Hence, this action research fieldwork aimed to achieve three goals:

(i)  To address existing problems in rural areas by empowering people to take the lead in finding solutions. This involves creating new opportunities and leveraging the capabilities of the entire community, with a particular emphasis on engaging young people and women.

(ii)  Through investments in infrastructure, technology, and education, the goal was to ensure access to basic services such as energy, water, sanitation, connectivity, and housing. Additionally, the aim was to create entrepreneurial ecosystems that not only help manage these services but also promote economic and social development in the community.

(iii)  In this sense, the primary objective was not only to reduce the gap between rural and urban areas but also to generate a "wave" of progress that ensures a constant improvement of rural spaces based on their own expectations. This approach involves actively involving the majority of social capital in the community and ensuring the sustainability of the environment.

Particularly, the SRC action research fieldwork focused on:

(i) Four key areas: Energy, Production, Education, and Technology.
(ii) It aimed to explore the interdependencies among these sectors through the use of solar panels, internet connection, and mobile phones.
(iii) The analysis of the experts started by understanding the habitus and behaviors of rural dwellers, rather than solely relying on existing software or systems.
(iv) A transdisciplinary team of experts collaborated with rural dwellers through the analysis conducted in Living Labs.
(v) Visual ethnography, interviews, and group dynamics were utilized to unpack the context of the site and facilitate collective visualization of the decision-making process.
(vi) Through this process, rural dwellers were empowered and it was suggested that the local team of experts could potentially continue to sustain the dynamic after the initial kick-off session.
(vii) Rural dwellers expressed surprise at the way in which the dynamics unfolded.
(vii) In both cases, it was suggested that this dynamic could be maintained as a Living Lab.

Thus, the SRC framework was deployed as a Living Lab with two general aims:

1. Improving the community 'hardware':
   a. The goal was to integrate the rural environment into global development processes by leveraging its territorial attributes. This involved providing appropriate technology, infrastructure, and services to address identified deficiencies and reduce gaps.
   b. Action research style: An action research approach was employed to identify specific technological and infrastructural needs of the rural environment. Through collaboration with local communities and stakeholders, the project worked to design and implement solutions that are tailored to their unique needs and circumstances.

2. Programming the necessary 'software' and its successive iterations:
   a. The aim was to design and facilitate processes that would enable the community hardware to fulfill its intended function, be sustainable, and contribute to long-term benefits. This involved fostering social innovation associated with assets and technology, which allowed for new forms of management, administration, execution, and the development of new instruments, tools, and combinations of factors aimed at improving social conditions.
   b. Action research style: To ensure the effectiveness and sustainability of the software, an action research approach was employed. This involved continuously assessing and improving its functionality based on the evolving needs and expectations of the community and environment. Ongoing collaboration with local communities and stakeholders facilitated the design and implementation of updates that are responsive to their changing needs and circumstances. Additionally, the software was developed through real-time prototyping with the community in Living Labs, as simulated during fieldwork.

It is important to note that the SRC project did not aim to compare the fieldwork action research in Bolivar (Colombia) and Pemba (Mozambique). Therefore, this project should not be considered a comparative research project. The fieldwork action research allowed the project to build the SRC model based on the insights gained from these two cases. It is crucial to consider this methodological distinction to avoid methodological nationalism and biases when modeling based on action research [64].

Nonetheless, since action research can be designed in various ways, leading to diverse outcomes, the interventions in Colombia and Mozambique considerably differed due to the SWOT analysis, which is presented in detail in this article:

**Table 1.** SWOT SRC Bolivar (Colombia).

| External | THREATS | OPPORTUNITIES |
| --- | --- | --- |
| | • Rural citizens had very little knowledge of other communities (which made SRC even more necessary).<br>• There was a significant gap (possibly indifference?) between the institutional world and reality, making it difficult to rely on government officials.<br>• There could be hesitancy towards SRC due to fear of losing control. | • The communities displayed a great variety and potential.<br>• The younger generation in all three communities showed enthusiasm and willingness to be involved and lead the transformation.<br>• Mobile telephones had great potential due to their widespread use.<br>• A small group of individuals could have a strong and positive catalytic effect, but it required delegation to persistent individuals |
| **Internal** | **WEAKNESSES** | **STRENGTHS** |
| | • Hardware experts were accustomed to working in silos within their own areas of expertise, which is common but not desirable.<br>• Grassroots organizations (CDS and Semana Foundation) had different approaches and varying levels of involvements in SRC. | • Internal leadership demonstrated vision and freedom of action.<br>• The operational manager exhibited excellent relational capacity and facilitation skills.<br>• Despite initial uncertainty and resistance, the hardware team members showed potential for collaboration after engaging with the communities during the fieldwork. It would have required an additional week to model the specifics.<br>• The internal team (software) had good logistical organization. |

**Table 2.** SWOT SRC Cabo Delgado (Mozambique).

| External | THREATS | OPPORTUNITIES |
| --- | --- | --- |
| | • There was a significant dispersion of communities, which made it challenging to establish and organized dynamic based on technical or territorial criteria.<br>• The institutional world was disconnected from the reality and lacked a clear vision for prioritization. They did not recognize the potential for endogenous community development through "Living Labs. | • There was no singular direction for development, although some individuals from different communities engaged in fruitful dialogue beyond their own community's development.<br>• There was a great diversity of development options available. However, it appeared that there was an imposed hierarchy favoring individuals with political rank and influence, which did not convince several people with the capacity for transformation.<br>• Education was identified as a clear need linked to production and entrepreneurship. The importance of connectivity emerged almost by accident, suggesting its potential as a triggering factor. |
| **Internal** | **WEAKNESSES** | **STRENGTHS** |
| | • The grassroots organization (Muleide) had good intentions but exhibited a paternalistic bias common among organizations in the area. With appropriate guidance and support, this bias could have been corrected, leading to interesting results. | • The internal/software team demonstrated complete integration and deep understanding of the field, showing openness to learning and experimentation.<br>• A work plan could have been developed for the aforementioned communities upon the completion of the fieldwork. |

Hence, after achieving the agreed-upon SWOT analysis results for both cases, policy recommendations were formulated to strategize SRC as a modeling framework. The results of the Living Labs in each case study, organized into four key areas (Energy, Production, Education, and Technology), provided evidence-based input for the SWOT analysis. The findings from the SWOT analysis led to the development of eight policy recommendations for SRC:

(i) Systemic sustainability: This involves a commitment to sustainability: economic, social, and environmental dimensions. It emphasizes the competitiveness and well-being of all actors and sectors in the territory, not just private or sectoral competitiveness.

(ii) Social cohesion: The goal is to improve the quality of life for all inhabitants of the territory, with specific actions targeting gender, childhood, youth, and indigenous populations. It aims to combat exclusion, poverty, inequality, and ensure the protection of human rights.

(iii) Territorial planning: It is necessary to contribute to ecologically sustainable, spatially harmonious, and socially equitable human development by organizing the use, exploitation, and occupation of the territory. This involves considering the needs of the population and incorporating recommendations generated by planning and management instruments.

(iv) Rural entrepreneurship: This component addresses the economic challenges faced by rural communities, particularly young people. It involves promoting technological, organizational, and management innovations, and creating "accompaniment ecosystems" that generate new productive and employment opportunities.

(v) Rural innovation: Sustainability relies on the ability to innovate, integrate existing knowledge within the territory, and leverage it for the common benefit. It requires continuous learning, adaptation, collaboration, networking, and effective information management.

(vi) Climate change resilience: Recognizing the need for coordinated responses to climate change impacts, mechanisms for planning and implementation must involve different levels of government and communities. Climate change resilience should be integrated into all public actions within the territory.

(vii) Rural–urban balance: Synergies between rural and urban territories should be established, moving away from viewing them as separate sectors. The aim is to foster a shared territorial network that addresses the common challenge of building sustainable, collaborative, and interconnected territories.

(viii) Governance: In intelligent, sustainable, collaborative, and inclusive territories, new forms of governance are created to ensure participatory and informed decision-making. Co-decision mechanisms unite the community around shared projects going beyond mere representation and involving input and evaluation from the entire community.

The following table presents the resulting SRC Living Lab model of intervention driven by action research. Table 3 includes long-term and mid-term impact, four objectives, sixteen results, and levers. Additionally, a process is defined as a template.

Table 3. SRC Living model of intervention. Long Term Impact, Mid Term Impact, Objectives, Results, and Levers.

| Long-Term Impact | Mid-Term Impact | OBJECTIVES | RESULTS | LEVERS |
|---|---|---|---|---|
| Reduce the development gap in rural areas through the use of information and knowledge exchange. | Create a sustainable living environment that enables access to personal and professional development opportunities. | Social Development | Improved access to sources of energy. | Low-voltage renewable energy network enabling household service |
| | | | Improved access to water and sanitation. | Potable water system using energy |
| | | | Improved health care services. | Provision and/or improvement of healthcare services using energy |
| | | | Improved quality education services. | Provision and/or improvement of education services using energy |
| | | Economic Development | Job creation. | Training program for green economy employment |
| | | | Improved employability skills. | Technical training for the provision of basic services |
| | | | Increased entrepreneurship. | Training and advisory plans for entrepreneurs |
| | | | Promotion of value chains. | Training, business plan advisory, and infrastructure provision |
| | | Environmental Development | Ensured waste management and reuse. | Design and implementation of a circular economy-based solid waste management system in the community |
| | | | Generated environmental risk management system. | Training and implementation of a risk management system that includes early warning |
| | | | Implemented monitoring and tracking system for natural resources. | Community organization and training to measure and record the evolution of natural resources |
| | | | Utilized environmental potential. | Study to determine possibilities for carrying out payment for environmental services projects |
| | | Community Development | Strengthened organizational structure. | Construction of an associative fabric that supports the management of different services/enterprises |
| | | | Generated participation channels. | Permanent consultation system taking advantage of technology. |
| | | | Increased knowledge of the needs and behavior of the population. | Design and use of a data platform that allows for the accumulation and analysis of population behaviors and the conclusions of participation processes. |
| | | | Reduced connectivity gap. | Installation of a home-use and production-unit internet connection system. |
| | | PROCESS | It is necessary to consider the initial elements of the project, which should start with high participation and a study that orders the territory and detects its needs/potential. | TERRITORIAL PLANNING |
| | | | | PARTICIPATION, LISTENING, AND CO-CREATION |
| | | | | PILOT SYSTEM |

## 5. Conclusions

To sum up, the article concludes with policy recommendations derived from the SRC action research fieldwork conducted in Mozambique and Colombia. In this final section, the article addresses the research question established at the beginning and provides case-specific policy recommendations. The research question was: How can the SRC framework serve as an international cooperation model that supports the SDGs? The article has presented the findings of the fieldwork action research, including a critical analysis of the hegemonic discourse surrounding SC, and has offered final recommendations. Furthermore, it has introduced the concept of SRC. In summary, the SRC framework has provided valuable insights for international cooperation models aiming to advance the SDGs in the Global South. The fieldwork action research conducted through the Living Labs, which involved the SWOT analysis and Modeling, has led to several conclusions that can be effectively utilized through the SRC approach to support the SDGs:

In Bolivar, Colombia, the following ideas have the potential to be implemented after discussion with stakeholders in the Living Lab, taking into account specific contextual conditions:

(i) Develop and implement smart tourism initiatives that promote and preserve Colombia's natural and cultural heritage while generating economic opportunities for rural communities.

(ii) Promote sustainable forestry practices and the development of bioeconomy initiatives to diversify and strengthen the rural economy.

(iii) Improve access to high-speed internet and digital infrastructure in rural areas to support the development of e-commerce, e-learning, and other digital services.

(iv) Implement smart waste management systems to reduce environmental pollution and enhance the health and well-being of rural communities.

(v) Foster social innovation and entrepreneurship among young people in rural and post-conflict areas by providing training, financing, and support for community-driven initiatives that address local challenges and create economic opportunities.

In Cabo Delgado, Mozambique, the following ideas have the potential to be implemented after discussion with stakeholders in the Living Lab, considering the specific contextual conditions:

(i) Develop and implement innovative and sustainable agricultural practices led by women through (data) cooperatives [74–76]. These practices should be tailored to the specific needs and challenges of rural communities in Mozambique such as introducing drought-resistant crops and utilizing precision agriculture techniques.

(ii) Expand access to renewable energy sources, such as solar and wind power, to enhance the energy security and economic opportunities of rural communities.

(iii) Develop and implement smart water management systems that enable rural communities to conserve and manage their water resources effectively, especially during periods of drought.

(iv) Establish community telecenters and other digital infrastructure to provide access to information and communication technologies (ICTs) and support the development of digital skills, literacy, and entrepreneurship in rural areas.

(v) Develop and implement smart transportation systems that improve the mobility and connectivity of rural communities including the adoption of electric vehicles and shared mobility solutions.

In summary, the SRC approach, implemented through Living Labs interventions using action research fieldwork, has demonstrated a highly positive impact on rural and remote communities in Colombia and Mozambique. The policy implications drawn from the SRC action research fieldwork in both countries indicate that investing in sustainable and innovative initiatives tailored to the specific needs and challenges of rural communities can yield positive outcomes for the environment and local economies. The Living Labs interventions, which employed an action research approach, showcased the potential of such initiatives to make a significant difference in remote and rural areas.

However, it is important to acknowledge the limitations of the fieldwork action research conducted in 2017, which took place before the COVID-19 pandemic [77]. The scalability and comprehensiveness of the intervention during the pre-pandemic times may need to be reconsidered in light of the increased vulnerability of rural environments in the post-pandemic era [5]. It is crucial to incorporate lessons learned from the pre-pandemic period and recognize the significance of integrating ICT into logistic and healthcare services to mitigate future global threats and align with the SDGs. SRC can proactively prepare rural and remote communities for such disruptions without being constrained by the hegemonic discourse surrounding SC and avoiding postcolonial implications [78,79].

**Funding:** This research was funded by (i) The Learned Society of Wales (LSW) under Grant 524205; (ii) Fulbright Scholar-In-Residence (S-I-R) Award 2022-23, Grant Number PS00334379 by US-UK Fulbright Commission and IIE, U.S. Department of State in California State University, Bakersfield; (iii) the Economic and Social Research Council (ESRC) under Grant ES/S012435/1 'WISERD Civil Society: Changing Perspectives on Civic Stratification/Repair'; (iv) Presidency of the Basque Government, External Affairs General Secretary, Basque Communities Abroad Direction; (v) the Economic and Social Research Council (ESRC) under Grant ES/M010996/1 'Urban Transformations'; (vi) Ayuda en Acción, Smart Rural Communities 2016-2019 Research & Knowledge Transfer; and (vii) Horizon-2020-Replicate-691735. The APC was funded by the Economic and Social Research Council (ESRC) under Grant ES/S012435/1 'WISERD Civil Society: Changing Perspectives on Civic Stratification/Repair' and the Economic and Social Research Council (ESRC) under Grant ES/M010996/1 'Urban Transformations'.

**Institutional Review Board Statement:** The study was conducted in accordance with the Central University Research Ethics Committee (CUREC) at the University of Oxford. This study obtained the written approval by the Research Ethics Approval on 3 July 2017. School of Anthropology and Museum Ethnography stated that 'the proposed research Smart Rural Communities has been judged as meeting appropriate ethical standards, and accordingly approval has been granted'. The reference number provided by the University of Oxford to this study was: SAME_C1A-17_079. Letter of Research Ethics Approval can be provided upon request to the author. Different forms were submitted and approved for the fieldwork in Colombia and Mozambique: (i) Information Sheet for Participants to Read (in Spanish and Portuguese); (ii) Oral Consent; (iii) Recording of Oral Consent; and (iv) Written Consent Form.

**Informed Consent Statement:** Informed consent was obtained from all subjects involved in the study and during fieldwork in Colombia and Mozambique. Central University Research Ethics Committee (CUREC) at the University of Oxford approved the Ethical forms, the action research procedure, and the informed consent statement as provided by the Principal Investigator, Dr Igor Calzada (https://www.ox.ac.uk/news-and-events/find-an-expert/dr-igor-calzada, https://profiles.cardiff.ac.uk/staff/calzadai and https://wiserd.ac.uk/people/igor-calzada/, accessed on 7 June 2023).

**Data Availability Statement:** Fieldwork data available upon request to the author.

**Acknowledgments:** The author would like to acknowledge the collaboration with the NGO Ayuda en Acción. The author is particularly grateful for the work in cooperation with Fernando Mudarra and Iban Askasibar, CEO and Alliances Director of the NGO, Ayuda en Acción. They took the leadership of this SRC project during the period 2016–2019 by culminating with the Summer School in 2019, St Sebastian, Basque Country (Spain). Equally in debt with the great rural dwellers that took responsibility of the action research fieldwork in Colombia and Mozambique. Obrigado to Josino Eugenio, the teacher at Pemba (Mozambique) for believing that kids' minds are the strongest technological weapons for the educational revolution.

**Conflicts of Interest:** The authors declare no conflict of interest.

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
