# Peer review of "Smart Rural Communities: Action Research in Colombia and Mozambique"

_sustainability, doi:10.3390/su15129521_

Round 1
Reviewer 1 Report
This study is judged to be a suggestive content that presents the results obtained through actual participation and observation of rural development programs focusing on linkage with the SDGs. Therefore, I offer the following comments.
1. It is necessary to supplement the grounds for selecting this research area under the SDG goals and to discuss the changes in the behavior of settlers before and after regional conflicts in more detail.
2. And the analytical process of the research tailored to the research problem is organized into 1) introduction, 2) literature review, 3) field survey presentation, 4) SWOT analysis, and 5) final policy recommendation. Since this system is a contextual structure that can be included in a policy report, it is necessary to change the research process to fit the table of contents of this journal in order to expand it to journal papers.
3. In addition, the policy implications of the survey results should be summarized limited to the contents presented in this study.
4. Prior research review should be organized around directly related contents to establish this research model.
5. So-called, mainly used in qualitative research. A conceptual definition of the Action Research Fieldwork and a process of schematization tailored to the design of this study should be presented. In other words, the analysis process and results can be trusted only when the scientific premise for social phenomena is presented in advance.
6. It is difficult to understand what the Living Lab presented in the Discussion in Chapter 4 was for, and through which process a common consensus was reached.
7. The SWOT analysis should also be expanded to discussions that can be strategized for each category, not simply limited to the analysis by four categories.
8. The formatting in Table 3 needs to be edited again.
9. The conclusion in Chapter 5 presents the main background, purpose, scope and analysis process of the study, and proposes a structure that suggests the limitations of the study and future tasks.
Author Response
Response to Reviewer 1 Comments
Review Report Form
Open Review
(x) I would not like to sign my review report
( ) I would like to sign my review report
Quality of English Language
( ) I am not qualified to assess the quality of English in this paper
( ) English very difficult to understand/incomprehensible
( ) Extensive editing of English language required
( ) Moderate editing of English language
( ) Minor editing of English language required
(x) English language fine. No issues detected
Yes |
Can be improved |
Must be improved |
Not applicable |
|
Is the content succinctly described and contextualized with respect to previous and present theoretical background and empirical research (if applicable) on the topic? |
( ) |
( ) |
(x) |
( ) |
Are all the cited references relevant to the research? |
( ) |
(x) |
( ) |
( ) |
Are the research design, questions, hypotheses and methods clearly stated? |
( ) |
( ) |
(x) |
( ) |
Are the arguments and discussion of findings coherent, balanced and compelling? |
( ) |
( ) |
(x) |
( ) |
For empirical research, are the results clearly presented? |
( ) |
( ) |
(x) |
( ) |
Is the article adequately referenced? |
( ) |
(x) |
( ) |
( ) |
Are the conclusions thoroughly supported by the results presented in the article or referenced in secondary literature? |
( ) |
( ) |
(x) |
( ) |
Comments and Suggestions for Authors
This study is judged to be a suggestive content that presents the results obtained through actual participation and observation of rural development programs focusing on linkage with the SDGs. Therefore, I offer the following comments.
- It is necessary to supplement the grounds for selecting this research area under the SDG goals and to discuss the changes in the behavior of settlers before and after regional conflicts in more detail.
Thank you for this comment. Both cases were selected by the NGO Ayuda en Acción given their interest in working with local rural dwellers in these two pre-and-post conflict areas respectively, Bolivar (Colombia) and Pemba (Mozambique). Thus, the selection of both cases was made by the leading organization of the project. I have provided the grounds for the selected case studies under the SDG goals. I have supplemented the general context by 2016 in Bolivar (Colombia) given the peace agreement dated by that time. The intervention of SRC took place in 2017. In addition, I have supplemented the general context after 2017, given that after the intervention in Pemba (Mozambique) several conflicts arose up to date. I have included two new maps as Figures, Figure 3 and Figure 4. And new long extract has been included. I believe now the article has been considerably improved after responding to this comment.
- And the analytical process of the research tailored to the research problem is organized into 1) introduction, 2) literature review, 3) field survey presentation, 4) SWOT analysis, and 5) final policy recommendation. Since this system is a contextual structure that can be included in a policy report, it is necessary to change the research process to fit the table of contents of this journal in order to expand it to journal papers.
Structure revised and changed by fitting into the journal style. I am afraid there is not any survey in this article as you indicated. The research is an action research design, and the content is clearly reflected upon the table of content of Sustainability article. In order to emphasise and even make this point clearer, Figure 2 has been designed and included in the article. As you can observe the action research design is made up of three steps: (i) State-of-the-Art: Literature Review; (ii) Fieldwork, and (iii) Modeling. A new Figure 2 has been included to show this sequence.
Accordingly, the article presents the following structure, which fits into the standard format of Sustainability: (i) Introduction; (ii) Literature Review: The State-of-The-Art on Smart Cities (SC), Smart Villages (SV), and Smart Rural Communities (SRC); (iii) Methods and Materials: Action Research Fieldwork in Colombia and Mozambique; (iv) Discussion: Findings and Policy Recommendations for Living Labs in SRC; and (v) Conclusions.
The action research fieldwork design has been deeply explained by providing new material: Figure 2, 3, 4, 5, and 6. I believe now the process is clear.
- In addition, the policy implications of the survey results should be summarized limited to the contents presented in this study.
Thank you. I am afraid but there is no survey in this article. Instead, the action research fieldwork included three methodological steps: Visual ethnography, in-depth interviews and Living labs. Actually, given the large amount of data gathered in both cases, this article did not include the material related to visual ethnography and in-depth interviews. The article elaborate in the findings after the Living Lab was established and the final session occurred (Figure 5 and 6). Again, there is no survey in this project and thus article. Regarding the policy recommendations (not implications), the article has remained as brief as possible by summarizing the content and the findings of this project. Actually, the reason why Tables 1, 2, and 3, were included respond to the fact to keep all brief and synthetic. Table 1 and 2 show SWOT in Colombia and Mozambique, while Table 3 shows the SRC Living model of intervention. This is the result of the Modeling as it presented in Figure 2.
Having said that, I have revised the discussion section in depth.
- Prior research review should be organized around directly related contents to establish this research model.
Thank you. Actually, prior research review is what the article refers to as ‘Literature Review: The State-of-The-Art on Smart Cities (SC), Smart Villages (SV), and Smart Rural Communities (SRC). This is the entire second section of the article in which I establish the research question around SRC as clearly state in the Abstract: The research question investigates how the Smart Rural Communities (SRC) framework can serve as an international cooperation model that supports the SDGs.
When you refer to ‘directly related contents’, in this article directly related contents are clearly addressed as: SC, SV, and SRC. They are presented as a solid literature background to establish the fieldwork action research design in the next section: Methods and Materials.
Having said that, I have reinforced the reference list including timely references in the topic addressed above.
- So-called, mainly used in qualitative research. A conceptual definition of the Action Research Fieldwork and a process of schematization tailored to the design of this study should be presented. In other words, the analysis process and results can be trusted only when the scientific premise for social phenomena is presented in advance.
Thank you. I agree with your point. Thus, I have included just at the beginning of the methodological section entitled ‘Methods and Materials: Action Research Fieldwork in Colombia and Mozambique’, a definition about Action Research and an entire paragraph. In addition to this, in order to show the Action Research design used through three steps (State-of-the-Art: Literature Review / Fieldwork / Modeling) Figure 2 has been included. Furthermore, to visualize the process Figure 3, 4, 5, and 6 illustrate the process, having included extracts, explanations, and additional nuanced methodological information about the action research fieldwork. I believe now the article has been improved.
- It is difficult to understand what the Living Lab presented in the Discussion in Chapter 4 was for, and through which process a common consensus was reached.
Thank you. The fourth section, Discussion, has been revised and a new paragraph has clarified how fieldwork action research proceeded to result in findings and policy recommendations. As I have included, SWOT analysis’ results was used to build common ground and consensus among stakeholders of this project SRC. Table 1 and 2 show SWOT analysis results.
- The SWOT analysis should also be expanded to discussions that can be strategized for each category, not simply limited to the analysis by four categories.
Thank you. I have connected SWOT with policy recommendations as you suggested. The way to strategize SWOT analysis findings/results (reached by consensus among all stakeholders of this project) was actually through the formulation of eight SRC policy recommendations.
I believe SWOT analysis should not be expanded insofar as it provides in a nutshell what we obtained from the fieldwork research in both cases. Thus, expanding more could be distracting in details readership. Policy recommendations are, as you kindly suggested, a way to strategize SWOT findings.
- The formatting in Table 3 needs to be edited again.
Thank you. I have edited Table 3.
- The conclusion in Chapter 5 presents the main background, purpose, scope and analysis process of the study, and proposes a structure that suggests the limitations of the study and future tasks.
Thank you for your kind comment. I have revised and included some ideas.
Submission Date
13 April 2023
Date of this review
15 Apr 2023 06:31:47
Reviewer 2 Report
The paper introduces mainly the results of a project, fieldwork that was carried out years ago. The concept, the approach to develop rural communities are acceptable but we all know that huge global changes have happened recently that had great influence on the potentials of smart developments, thus results from 2017 are not relevant. It is understandable that the paper wants to focus on concrete project and fieldwork but in 2023, mroe recent results, experiments are needed to be relevant, especially when we come to strategies, future development directions for rural areas.
Author Response
Response to Reviewer 2 Comments
Open Review
(x) I would not like to sign my review report
( ) I would like to sign my review report
Quality of English Language
( ) I am not qualified to assess the quality of English in this paper
( ) English very difficult to understand/incomprehensible
( ) Extensive editing of English language required
( ) Moderate editing of English language
( ) Minor editing of English language required
(x) English language fine. No issues detected
Yes |
Can be improved |
Must be improved |
Not applicable |
|
Is the content succinctly described and contextualized with respect to previous and present theoretical background and empirical research (if applicable) on the topic? |
( ) |
(x) |
( ) |
( ) |
Are all the cited references relevant to the research? |
(x) |
( ) |
( ) |
( ) |
Are the research design, questions, hypotheses and methods clearly stated? |
( ) |
(x) |
( ) |
( ) |
Are the arguments and discussion of findings coherent, balanced and compelling? |
( ) |
(x) |
( ) |
( ) |
For empirical research, are the results clearly presented? |
( ) |
(x) |
( ) |
( ) |
Is the article adequately referenced? |
(x) |
( ) |
( ) |
( ) |
Are the conclusions thoroughly supported by the results presented in the article or referenced in secondary literature? |
( ) |
(x) |
( ) |
( ) |
Comments and Suggestions for Authors
The paper introduces mainly the results of a project, fieldwork that was carried out years ago. The concept, the approach to develop rural communities are acceptable but we all know that huge global changes have happened recently that had great influence on the potentials of smart developments, thus results from 2017 are not relevant. It is understandable that the paper wants to focus on concrete project and fieldwork but in 2023, mroe recent results, experiments are needed to be relevant, especially when we come to strategies, future development directions for rural areas.
Thank you for considering my article interesting and acceptable for publication. I entirely agree with the massive disruption provoked by the pandemics. Actually, I have included several new references in this topic and particularly an article that is the introduction of a Special Issue in the journal Citizenship Studies that exactly addressed the pre-post-pandemic transformations: Digital Citizenship in the Post-Pandemic Urban Realm (https://www.tandfonline.com/toc/ccst20/27/2). I am the guest editor of such Special Issue. Thus, I have incorporated several ideas about this important fact in relation to SRC. Actually, there are several new extracts addressing the grounds of the two cases, Bolivar in Colombia and Pemba in Mozambique, in pre- and post-pandemic times.
The article aims to present the intense fieldwork research carried out in 2017. So the analysis is retrospective and by including the pre and post-pandemic sequence, the analysis and fieldwork research remain clearly valid. We all know pandemic has created an entirely new environment, hitting the poor and the remote/rural the most. Actually, my latest book revolved around the postpandemics (technological democracies): Emerging Digital Citizenship Regimes (Emerald, 2022). So, you have giving me the suggestion to include the reference and elaborate a bit on thesis. The fieldwork conducted in 2017 is actually data and information collection originated in 2017. But, anthropologist tend to keep data for decades and then the analyses it. Otherwise, ethnographic and anthropological data would not remain valid as such, which clearly is not the case. Data collection and data interpretation could easily happen with a gap during a fieldwork process, which essentially does not invalidate the data or even the conclusion. The point is that the context should be clearly describe or interpret. Essentially what I did in this revised version. Thank you for your comment and for helping me improving my article.
Here new extracts contextualizing your comment in my article:
This article aims to contribute to the ongoing discussion on rural development programs and their alignment with the SDGs in the Global South, both before and after the pandemic. By examining the successes, limitations, and lessons learned from existing initiatives, it seeks to shed light on effective strategies for addressing the multifaceted challenges faced by rural communities. Furthermore, although the fieldwork action research was conducted in 2017, before the pandemic, the article aims to retrospectively reflect on the impact of the COVID-19 pandemic on rural development efforts, considering the findings and evidence from the fieldwork action research conducted in Colombia and Mozambique.
COVID-19 was spreading rapidly, and its tragic aftermath showed that the world is highly interconnected. Acknowledging the particularities of the Global South in relation to the Global North is necessary to solve a great number of problems [28]. Shockingly, COVID-19 made all world citizens pandemic citizens, sharing the same fear, uncertainty, and risks regardless of their location in the world [29-31]. However, it was unlikely that the pandemic crisis and its algorithmic disruptive vulnerabilities equally affected citizens in the Global South and the Global North. It became evident that the pandemic crisis forced the world into an algorithmic crisis, in which citizens' data could be used for unfair or unethical purposes by governments or private companies. The proliferation of new emerging digitalization/datafication apps, including ChatGPT and Metaverse among others, only served to confirm this early intuition. Above all, and while considering the digital risks, Living Labs are among the various resilience strategies worth considering to address the aftermath of COVID-19 through social innovation [32-34].
However, it is important to acknowledge the limitations of the fieldwork action research conducted in 2017, which took place before the COVID-19 pandemic [77]. The scalability and comprehensiveness of the intervention during the pre-pandemic times may need to be reconsidered in light of the increased vulnerability of rural environments in the post-pandemic era [5]. It is crucial to incorporate lessons learned from the pre-pandemic period and recognize the significance of integrating ICT into logistic and healthcare services to mitigate future global threats and align with the SDGs. SRC can proactively prepare rural and remote communities for such disruptions without being constrained by the hegemonic discourse surrounding SC and avoiding postcolonial implications [78-79].
Submission Date
13 April 2023
Date of this review
26 Apr 2023 11:27:28
Reviewer 3 Report
1. The abstract does not clearly explain the results of the research
2. Introduction has not conveyed the problem, and the theory referred to, so there is a missing link
3. The problem statement and state-of-the-art do not yet convey the theoretical background
4. The method does not convey the key variables that should be explained, so the relationship between them isn't clear
5. The research results do not yet answer the aims of the research.
6. The conclusion reads more like an extension of the discussion. The conclusion is that the initial tests give evidence that the proposed treatment merits further investigation
7. References have not shown a good state of the art. It should be updated so that the research novelties are visible
8. The article is not yet well elaborated—too much redundant information in the introduction, as well as the results and discussion part.
9. you have to explain how the concrete steps of action research
10. There is an explanation before and after the activity
The English writing needs a lot of improvement
Author Response
Response to Reviewer 3 Comments
Open Review
( ) I would not like to sign my review report
(x) I would like to sign my review report
Quality of English Language
( ) I am not qualified to assess the quality of English in this paper
( ) English very difficult to understand/incomprehensible
(x) Extensive editing of English language required
( ) Moderate editing of English language
( ) Minor editing of English language required
( ) English language fine. No issues detected
Yes |
Can be improved |
Must be improved |
Not applicable |
|
Is the content succinctly described and contextualized with respect to previous and present theoretical background and empirical research (if applicable) on the topic? |
( ) |
( ) |
( ) |
(x) |
Are all the cited references relevant to the research? |
( ) |
( ) |
(x) |
( ) |
Are the research design, questions, hypotheses and methods clearly stated? |
( ) |
( ) |
(x) |
( ) |
Are the arguments and discussion of findings coherent, balanced and compelling? |
( ) |
( ) |
(x) |
( ) |
For empirical research, are the results clearly presented? |
( ) |
( ) |
(x) |
( ) |
Is the article adequately referenced? |
( ) |
( ) |
(x) |
( ) |
Are the conclusions thoroughly supported by the results presented in the article or referenced in secondary literature? |
( ) |
( ) |
(x) |
( ) |
Comments and Suggestions for Authors
- The abstract does not clearly explain the results of the research
The Abstract has been revised accordingly. The results of the research are stated as follows:
This article contributes to the ongoing discussion on rural development programs aligning with the Sustainable Development Goals (SDGs) in the Global South. The research question examines how the Smart Rural Communities (SRC) framework can support the SDGs as an international cooperation model. The article presents findings from fieldwork action research, including a critical analysis of the hegemonic discourse on smart cities, and provides final recommendations. Additionally, it introduces the concept of SRC. The fieldwork action research was conducted in post-conflict rural areas in Colombia's Bolivar region and remote settlements in Mozambique's Cabo Delgado province. Led by Ayuda En Acción, in collaboration with co-operatives such as Mundukide Foundation, Alecop, and LKS from Mondragon Co-operative Corporation, these interventions aimed to engage local communities through Living Labs. They utilized Information and Communication Technologies (ICT) and social innovation to promote the well-being of rural residents. The article comprises (i) an introduction, (ii) literature review, (iii) presentation of the fieldwork action research in Colombia and Mozambique, (iv) findings from a SWOT analysis and policy recommendations for SRC Living Labs, and (v) conclusions addressing the research question. The SRC framework offers valuable insights for international cooperation models striving to achieve the SDGs in the Global South.
- Introduction has not conveyed the problem, and the theory referred to, so there is a missing link
Thanks. I have proceeded as follows in order to respond to this point: The introduction has been rewritten entirely to respond to this comment. The introduction frames now the research question and the ‘theory’ or better said, the literature review. There is not such a missing link given the introduction covers the research question while introducing literature review.
I believe now, the introduction frames the article and convey the research question (rather than the problem). In the final section, Conclusions, actually, I explicitly responded to the research question. The new introduction sheds light on the rationale behind this article. Challenges have been deeply addressed to show a plethora of ongoing challenges. The introduction section now has been renamed as: Introduction: Smart Rural Communities and SDGs in the Global South.
- The problem statement and state-of-the-art do not yet convey the theoretical background
Thank you for your observation. The section Literature review now has been renamed as ‘Literature Review: The State-of-the-Art on Smart Cities (SC), Smart Villages (SV), and Smart Rural Communities (SRC). While the introduction has been extended to connect with the state-of-the-art, the Literature review equally has included an introductory paragraph that convey with the theoretical background. I believe now this link works.
The method does not convey the key variables that should be explained, so the relationship between them isn't clear
The section Methodology has been renamed as ‘Methods and Materials: Action Research Fieldwork in Colombia and Mozambique’. There are no variables insofar as this is not quantitative research. This section explained deeply the fieldwork action research design. Actually, Figure 2, explains the action research design and how modeling has been reached out after the state-of-the-art/Literature Review and the Fieldwork. In addition to this, Figure 3, 4, 5, and 6 have been included to show properly the way the method has been put into practice. Action research definitions also have been included and the connection with the section 4 Discussion has been ensured too.
- The research results do not yet answer the aims of the research.
Thank you. Discussion section and Conclusion sections show the respond to your point. Research results are gradually presented and through Table 1, 2, (SWOT) and 3 (Modeling). Before the end of Discussion section, eight policy recommendations of SRC are presented.
The abstract and the article clearly specify the aim and the research question of this article:
The research question investigates how the Smart Rural Communities (SRC) framework can serve as an international cooperation model that supports the SDGs. The article presents the findings of fieldwork action research, including a critical analysis of the hegemonic discourse around smart cities, and final recommendations.
The conclusion section responds to the research question by presenting the policy recommendations for each case.
- The conclusion reads more like an extension of the discussion. The conclusion is that the initial tests give evidence that the proposed treatment merits further investigation
The conclusion section has incorporated several elements to respond to your comment. Indeed, the conclusions are an extension of the discussion section as I argued in the previous point. The article is presented gradually as a sequence insofar as it has been guided by action research. Probably, your way of reading the article is linear and action research is an iterative process that I present throughout the article.
Indeed, the initial research question is responded for both cases in light of the fieldwork findings presented through SWOT analysis and Modeling.
- References have not shown a good state of the art. It should be updated so that the research novelties are visible
A large list of recent and timely references has been included with cutting-edge research and novelties as follows:
- Calzada, I. Smart City Citizenship. Elsevier: Amsterdam, 2021.
- Anastasiou, E.; Manika, S.; Ragazou, K.; Katsios, I. Territorial and Human Geography Challenges: How Can Smart Villages Support Rural Development and Population Inclusion? Social Sciences. 2021, 10, 193. https://doi.org/10.3390/socsci10060193
- Kalinowski, S.; Komorowski, L.; Rosa, A. The Smart Village Concept: Examples from Poland. Instytut Rozwoju Wsi i Rolnictwa PAN/Institute of Rural and Agricultural Development: Warszawa, 2022. ISBN: 978-83-961048-1-6. DOI:10.53098/978-83-961048-1-6. Available online https://www.researchgate.net/publication/360963908_THE_SMART_VILLAGE_CONCEPT_EXAMPLES_FROM_POLAND [Accessed: 2023-05-10].
- Stojanova, S.; Lentini, G.; Niederer, P.; Egger, T.; Cvar, N.; Kos, A.; Duh, E.S. Smart Villages Policies: Past, Present and Future. Sustainability, 2021, 13, DOI:10.3390/su13041663.
- Anastasiou, E.; Manika, S.; Ragazou, K.; Katsios, I. Territorial and Human Geography Challenges: How Can Smart Villages Support Rural Development and Population Inclusion? Social Sciences, 2021, 10, DOI:10.3390/socsci10060193.
- Gerli, P.; Navio Marco, J.; Whalley, J. What Makes a Smart Village Smart? A Review of the Literature. Transforming Government: People, Process and Policy, 2022, 16(3), 292-304. DOI:10.1108/TG-07-2021-0126.
- Komorowski, L.; Stanny, M. Smart Villages: Where Can They Happen? Land, 2020, 9, 151. DOI:10.3390/land9050151.
- Wolski, O.; Wojcik, M. Smart Villages Revisited: Conceptual Background and New Challenges at the Local Level. In A. Visvizi, M.D. Lytras & G. Mudri (Eds), Smart Villages in the EU and Beyond. pp. 29-48. Bingley: Emerald.
- Degada, A., Thapliyal, H., Mohanty, S.P. Smart Village: An IoT Based Digital Transformation. In 2021 IEEE 7th World Forum on Internet of Things (WF-IoT). pp. 459-463.
- Morozov, E. To Save Everything, Click Here: The Folly Technological Solutionism. PublicAffairs: NYC, 2014.
- European Network for Rural Development, Connecting Rural Europe. Available online: https://ec.europa.eu/enrd/smart-and-competitive-rural-areas/smart-villages/smart-villages-portal_en.html [Accessed: 2023-05-15].
- Visvizi, A.; Lytras, M.D.; Mudri, G. Smart Villages in the EU and Beyond. Emerald: Bingley, 2019.
- Rahoveanu, M.M.R.; Serban, V.; Zugravu, A.G.; Rahoveanu, A.T.; Cristea, D.S.; Nechita, P.; Simionescu, C.S. Perspectives on Smart Villages from a Bibliometric Approach. Sustainability, 2022, 14, DOI:10.3390/su141710723.
- Wimmer, A.; Glick Schiller, N. Methodological nationalism and beyond: Nation-state building, migration and the social sciences. Global Networks, 2002, 2(4), 301-334. DOI: 10.1111/1471-0374.00043.
- Datta, A. New urban utopias of postcolonial India: Entrepreneurial urbanization in Dholera smart city, Gujarat. Dialogues in Human Geography, 2015, 5(1), 3-22. DOI:10.1177/2043820614565748.
- Jazeel, T. Mainstreaming geography's decolonial imperative. Transactions of the Institute of British Geographers, 2017, 42(3), 334-337. DOI:10.1111/tran.12200.
- Calzada, I.; Cowie, P. Beyond smart and data-driven city-regions? Rethinking stakeholder-helixes strategies. Regions, 2017, 308(4), 25-28. DOI:10.1080/13673882.2017.11958675.
- Calzada, I.; Chautón, A.; Di Siena, D. (2013), MacroMesoMicro: Systemic Territory Framework from the perspective of Social Innovation. ISBN: 978-84-616-5217-4.
- Calzada, I. How do small nations cooperate? Wales and the Basque Country. Regional Studies, Regional Sciences. 2023.
- Soeiro, D. Smart cities and innovative governance systems: a reflection on urban living labs and action research. Fennia, 2021, 199(1). DOI:10.11143/fennia.97054.
- Thees, H.; Pechlaner, H.; Olbrich, N.; Schuhbert, A. The Living Lab as a tool to promote residents' participation in destination governance. Sustainability, 2020, 12(3). DOI:10.3390/su12031120.
- Zavratnik, V.; Superina, A.; Stojmenova Duh, E. Living labs for rural areas: Contextualization on living lab frameworks, concepts, and practices. Sustainability, 2019, 11(14), 3797. DOI:10.3390/su11143797.
- Calzada, I. Local entrepreneurship through a multistakeholders' tourism living lab in the post-violence/peripheral era in the Basque Country. Regional Science Policy & Practice, 2019, 11(3), 451-466. DOI:10.1111/rsp3.12130.
- Bühler, M.M.; Calzada, I.; Cane, I.; Jelinek, T.; Kapoor, A.; Mannan, M.; Mehta, S.; Micheli, M.; Mookerje, V.; Nübel, K.; Pentland, A.; Scholz, T.; Siddarth, D.; Tait, J.; Vaitla, B.; Zhu, J., Data Cooperatives as Catalysts for Collaboration, Data Sharing, and the (Trans)Formation of the Digital Commons. org, 2023, 2023040130. DOI:10.20944/preprints202304.0130.v1.
- Calzada, I. Data Co-operatives through Data Sovereignty. Smart Cities, 2021, 4, 1158–1172. https://doi.org/10.3390/smartcities4030062
- Calzada, I. Platform and Data Co-operatives Amidst European Pandemic Citizenship, Sustainability, 2020, 12(20): 8309. DOI: 10.3390/su12208309.
- Katz, I.T.; Weintraub, R.; Bekker, L.-G.; Brandt, A.M. From Vaccine Nationalism to Vaccine Equity: Finding a Path Forward. New England Journal of Medicine. DOI:10.1056/NEJMP2103614.
- Calzada, I. Emerging Digital Citizenship Regimes: Postpandemic Technopolitical Democracies. Emerald: Bingley, 2022.
- Caragliu, A.; Del Bo, C.F. Smart innovative cities: The impact of smart city policies on urban innovation. Technological Forecasting & Social Change, 2019, 143. DOI:10.1016/j.techfore.2018.07.022.
- The article is not yet well elaborated—too much redundant information in the introduction, as well as the results and discussion part.
Thank you. The introduction and literature review section has been entirely written from scratch by removing all redundancies. Discussion and conclusions section have been better linked. Thus, the article can be followed through a sequence. The article has been revised section by section.
- you have to explain how the concrete steps of action research
Thank you. I have included Figure 2 by which the action research steps can be seen clearly. In the text, I have further elaborated these explanations. Action research has been defined and contextualized. I believe now the article has improved by responding to your comment.
- There is an explanation before and after the activity
Explanations have been added throughout the whole article.
Comments on the Quality of English Language
The English writing needs a lot of improvement
The article has been revised and entirely proofread accordingly. Thanks
Submission Date
13 April 2023
Date of this review
05 May 2023 03:43:58
Reviewer 4 Report
Research on the concepts of smart villages and smart communities is extremely important. Why? Because they show that the countryside can be a better place to live. That the level and quality of life in the countryside can improve. Therefore, it is gratifying to take up the topic.
However, I have a few comments:
1. insufficiently recognized literature on the subject (important items are missing, e.g. Kalinowski et al. The smart village concept..., Wójcik, Komorowski or Stojanova.
2. lack of attention to the fact that the concept refers to improving the quality of life in the countryside. The very idea of SV is not to look for digital-communication solutions but to create conditions for a good life and greater activity of the rural population (e.g. Kalinowski).
3 It is also worth noting that the concept itself has been implemented for a short time and is linked to other concepts - in the EU, for example, it is village renewal, but it is also derived from the concept of sustainable development.
Author Response
Response to Reviewer 4 Comments
Open Review
( ) I would not like to sign my review report
(x) I would like to sign my review report
Quality of English Language
(x) I am not qualified to assess the quality of English in this paper
( ) English very difficult to understand/incomprehensible
( ) Extensive editing of English language required
( ) Moderate editing of English language
( ) Minor editing of English language required
( ) English language fine. No issues detected
Yes |
Can be improved |
Must be improved |
Not applicable |
|
Is the content succinctly described and contextualized with respect to previous and present theoretical background and empirical research (if applicable) on the topic? |
( ) |
( ) |
(x) |
( ) |
Are all the cited references relevant to the research? |
( ) |
(x) |
( ) |
( ) |
Are the research design, questions, hypotheses and methods clearly stated? |
( ) |
(x) |
( ) |
( ) |
Are the arguments and discussion of findings coherent, balanced and compelling? |
( ) |
(x) |
( ) |
( ) |
For empirical research, are the results clearly presented? |
(x) |
( ) |
( ) |
( ) |
Is the article adequately referenced? |
( ) |
( ) |
(x) |
( ) |
Are the conclusions thoroughly supported by the results presented in the article or referenced in secondary literature? |
( ) |
(x) |
( ) |
( ) |
Comments and Suggestions for Authors
Research on the concepts of smart villages and smart communities is extremely important. Why? Because they show that the countryside can be a better place to live. That the level and quality of life in the countryside can improve. Therefore, it is gratifying to take up the topic.
However, I have a few comments:
- insufficiently recognized literature on the subject (important items are missing, e.g. Kalinowski et al. The smart village concept..., Wójcik, Komorowski or Stojanova.
Thank you very much. This is a necessary observation. Actually, I have included a broad list of references following your suggestion.
- Calzada, I. Smart City Citizenship. Elsevier: Amsterdam, 2021.
- Anastasiou, E.; Manika, S.; Ragazou, K.; Katsios, I. Territorial and Human Geography Challenges: How Can Smart Villages Support Rural Development and Population Inclusion? Social Sciences. 2021, 10, 193. https://doi.org/10.3390/socsci10060193
- Kalinowski, S.; Komorowski, L.; Rosa, A. The Smart Village Concept: Examples from Poland. Instytut Rozwoju Wsi i Rolnictwa PAN/Institute of Rural and Agricultural Development: Warszawa, 2022. ISBN: 978-83-961048-1-6. DOI:10.53098/978-83-961048-1-6. Available online https://www.researchgate.net/publication/360963908_THE_SMART_VILLAGE_CONCEPT_EXAMPLES_FROM_POLAND [Accessed: 2023-05-10].
- Stojanova, S.; Lentini, G.; Niederer, P.; Egger, T.; Cvar, N.; Kos, A.; Duh, E.S. Smart Villages Policies: Past, Present and Future. Sustainability, 2021, 13, DOI:10.3390/su13041663.
- Anastasiou, E.; Manika, S.; Ragazou, K.; Katsios, I. Territorial and Human Geography Challenges: How Can Smart Villages Support Rural Development and Population Inclusion? Social Sciences, 2021, 10, DOI:10.3390/socsci10060193.
- Gerli, P.; Navio Marco, J.; Whalley, J. What Makes a Smart Village Smart? A Review of the Literature. Transforming Government: People, Process and Policy, 2022, 16(3), 292-304. DOI:10.1108/TG-07-2021-0126.
- Komorowski, L.; Stanny, M. Smart Villages: Where Can They Happen? Land, 2020, 9, 151. DOI:10.3390/land9050151.
- Wolski, O.; Wojcik, M. Smart Villages Revisited: Conceptual Background and New Challenges at the Local Level. In A. Visvizi, M.D. Lytras & G. Mudri (Eds), Smart Villages in the EU and Beyond. pp. 29-48. Bingley: Emerald.
- Degada, A., Thapliyal, H., Mohanty, S.P. Smart Village: An IoT Based Digital Transformation. In 2021 IEEE 7th World Forum on Internet of Things (WF-IoT). pp. 459-463.
- Morozov, E. To Save Everything, Click Here: The Folly Technological Solutionism. PublicAffairs: NYC, 2014.
- European Network for Rural Development, Connecting Rural Europe. Available online: https://ec.europa.eu/enrd/smart-and-competitive-rural-areas/smart-villages/smart-villages-portal_en.html [Accessed: 2023-05-15].
- Visvizi, A.; Lytras, M.D.; Mudri, G. Smart Villages in the EU and Beyond. Emerald: Bingley, 2019.
- Rahoveanu, M.M.R.; Serban, V.; Zugravu, A.G.; Rahoveanu, A.T.; Cristea, D.S.; Nechita, P.; Simionescu, C.S. Perspectives on Smart Villages from a Bibliometric Approach. Sustainability, 2022, 14, DOI:10.3390/su141710723.
- Wimmer, A.; Glick Schiller, N. Methodological nationalism and beyond: Nation-state building, migration and the social sciences. Global Networks, 2002, 2(4), 301-334. DOI: 10.1111/1471-0374.00043.
- Datta, A. New urban utopias of postcolonial India: Entrepreneurial urbanization in Dholera smart city, Gujarat. Dialogues in Human Geography, 2015, 5(1), 3-22. DOI:10.1177/2043820614565748.
- Jazeel, T. Mainstreaming geography's decolonial imperative. Transactions of the Institute of British Geographers, 2017, 42(3), 334-337. DOI:10.1111/tran.12200.
- Calzada, I.; Cowie, P. Beyond smart and data-driven city-regions? Rethinking stakeholder-helixes strategies. Regions, 2017, 308(4), 25-28. DOI:10.1080/13673882.2017.11958675.
- Calzada, I.; Chautón, A.; Di Siena, D. (2013), MacroMesoMicro: Systemic Territory Framework from the perspective of Social Innovation. ISBN: 978-84-616-5217-4.
- Calzada, I. How do small nations cooperate? Wales and the Basque Country. Regional Studies, Regional Sciences. 2023.
- Soeiro, D. Smart cities and innovative governance systems: a reflection on urban living labs and action research. Fennia, 2021, 199(1). DOI:10.11143/fennia.97054.
- Thees, H.; Pechlaner, H.; Olbrich, N.; Schuhbert, A. The Living Lab as a tool to promote residents' participation in destination governance. Sustainability, 2020, 12(3). DOI:10.3390/su12031120.
- Zavratnik, V.; Superina, A.; Stojmenova Duh, E. Living labs for rural areas: Contextualization on living lab frameworks, concepts, and practices. Sustainability, 2019, 11(14), 3797. DOI:10.3390/su11143797.
- Calzada, I. Local entrepreneurship through a multistakeholders' tourism living lab in the post-violence/peripheral era in the Basque Country. Regional Science Policy & Practice, 2019, 11(3), 451-466. DOI:10.1111/rsp3.12130.
- Bühler, M.M.; Calzada, I.; Cane, I.; Jelinek, T.; Kapoor, A.; Mannan, M.; Mehta, S.; Micheli, M.; Mookerje, V.; Nübel, K.; Pentland, A.; Scholz, T.; Siddarth, D.; Tait, J.; Vaitla, B.; Zhu, J., Data Cooperatives as Catalysts for Collaboration, Data Sharing, and the (Trans)Formation of the Digital Commons. org, 2023, 2023040130. DOI:10.20944/preprints202304.0130.v1.
- Calzada, I. Data Co-operatives through Data Sovereignty. Smart Cities, 2021, 4, 1158–1172. https://doi.org/10.3390/smartcities4030062
- Calzada, I. Platform and Data Co-operatives Amidst European Pandemic Citizenship, Sustainability, 2020, 12(20): 8309. DOI: 10.3390/su12208309.
- Katz, I.T.; Weintraub, R.; Bekker, L.-G.; Brandt, A.M. From Vaccine Nationalism to Vaccine Equity: Finding a Path Forward. New England Journal of Medicine. DOI:10.1056/NEJMP2103614.
- Calzada, I. Emerging Digital Citizenship Regimes: Postpandemic Technopolitical Democracies. Emerald: Bingley, 2022.
- Caragliu, A.; Del Bo, C.F. Smart innovative cities: The impact of smart city policies on urban innovation. Technological Forecasting & Social Change, 2019, 143. DOI:10.1016/j.techfore.2018.07.022.
- lack of attention to the fact that the concept refers to improving the quality of life in the countryside. The very idea of SV is not to look for digital-communication solutions but to create conditions for a good life and greater activity of the rural population (e.g. Kalinowski).
This is an interesting interpretation. Thank you very much. Indeed, I agree with this point. I have included a whole section on this. Actually, SV and SRC share the same approach while SRC suggests emulating living ecosystems through Living Labs.
3 It is also worth noting that the concept itself has been implemented for a short time and is linked to other concepts - in the EU, for example, it is village renewal, but it is also derived from the concept of sustainable development.
This is a relevant point. Thank you very much it is very pertinent. As such, I have introduced two new paragraphs highlighting this idea and reinforcing it with new references that revolve around the EU resonance and need for better tailored tools and instruments. These new extracts show the way action research through Living Labs could be a way to overcome the limitations presented by the approach Smart Villages.
Submission Date
13 April 2023
Date of this review
24 Apr 2023 12:57:56
Round 2
Reviewer 1 Report
Unfortunately, there are still many deficiencies for publication in this journal.
Although the contributor diligently revised the paper, it is judged that it is still difficult to overcome the limitations of qualitative research.
It is also difficult to trust the conceptual definition of Action Research Fieldwork and the process of schematization tailored to this research design.
In particular, the biggest point is that the content discrimination, such as the research process and journal purpose, is still insufficient to bring the context of the policy report researched by the contributor into the paper of this society.
The development of the article is not logical, and careful word selection is required in this journal.
Reviewer 3 Report
the revision has been carried out properly, all input has been corrected according to the suggestions given